

# Metabolic profile alterations in juvenile rats with bladder overactivity induced by short-term high-fructose intake

Xu Cui, Zhiqiang Chen, Longyao Xu, Changwei Wu and Chaoming Zhou

College of Clinical Medicine for Obstetrics & Gynecology and Pediatrics, Fujian Medical University, Fuzhou, China
Fujian Children's Hospital (Fujian Branch of Shanghai Children's Medical Center), Fuzhou, China

## ABSTRACT

**Background**. Overactive bladder syndrome (OAB) is a common condition that affects both adults and children, often causing significant discomfort and negatively impacting quality of life. This study aimed to preliminarily evaluate whether fructose exposure induces OAB-like symptoms in juvenile rats and to explore potential metabolic mechanisms involved.

**Methods**. Three-week-old female Sprague-Dawley rats were randomly assigned to two groups: a model group fed a 60% fructose diet for one week, and a control group fed a standard diet. Body weight, blood biochemical parameters, urination behavior, and urodynamic function were assessed. Bladder tissue was analyzed using hematoxylin and eosin staining, immunohistochemistry, and immunofluorescence. Molecular and metabolic changes were evaluated via Western blot and metabolomic analysis.

**Results**. Short-term high fructose intake led to significant metabolic and functional changes in juvenile rats. The model group showed reduced body weight and altered lipid metabolism, while glucose and insulin levels remained largely unchanged. Urodynamic assessments revealed increased non-voiding contractions, shortened intervals between contractions, and impaired bladder storage function. Open field tests indicated increased urinary frequency and open field test results. Histological analysis demonstrated disorganized bladder smooth muscle structure and elevated oxidative stress. Metabolomic profiling revealed marked alterations in energy metabolism, including enhanced glycolysis, pentose phosphate pathway activation, and accumulation of ATP and lactate. Western blot analysis confirmed activation of the CaMKK2/AMPK signaling pathway in bladder tissue.

**Conclusions**. Short-term high-fructose intake induces OAB-like symptoms in juvenile rats, accompanied by fructose toxicity, oxidative stress, calcium ion accumulation, and activation of the CaMKK2/AMPK signaling pathway. These findings highlight a potential link between fructose metabolism and bladder dysfunction, offering new insights for exploring metabolic or antioxidant-based therapeutic strategies in pediatric OAB.

Corresponding author
Chaoming Zhou,
fjsetyyzcm@163.com

# INTRODUCTION

Overactive bladder syndrome (OAB) is a common urological condition characterized by urinary urgency, increased frequency, and incontinence in the absence of infection or identifiable neurological disease, as defined by the International Continence Society (ICS) (*Abrams et al., 2002*). OAB affects both adults and children, with a prevalence of 9%–12% in the pediatric population (*Xing et al., 2020*). While the pathophysiology of OAB remains incompletely understood, proposed risk factors in children include constipation, delayed toilet training, urinary tract infections, obesity, and family history (*Cardoso et al., 2021*). OAB significantly impacts the quality of life for both children and their families, often leading to emotional distress, behavioral issues, and social withdrawal (*Dai et al., 2024*; *Özçift & Micoogullari, 2022*). Diagnosis is primarily based on clinical evaluation, with non-invasive tools such as the overactivity index and uroflowmetry commonly employed as substitutes for urodynamic studies (*Abdovic et al., 2022*). The management of pediatric OAB typically follows a stepwise approach, beginning with standard urotherapy that includes timed voiding, fluid management, and behavioral interventions (*Shim & Oh, 2023*). When conservative treatment fails, pharmacotherapy is employed, most commonly using antimuscarinic agents such as oxybutynin or tolterodine, although only a limited number of these drugs are officially approved for use in children (*Ramsay, Lapointe & Bolduc, 2022*). Antimuscarinics are often associated with adverse effects (*e.g.*, dry mouth, constipation, cognitive disturbances), and discontinuation rates remain high (*Kim et al., 2022*). While β3-adrenoceptor agonists offer improved tolerability, pediatric clinical data remain limited, especially for long-term safety and efficacy (*Krhut et al., 2022*; *Michel et al., 2023*).

In recent years, the role of nutrition in the development and management of OAB has gained increasing attention. Clinical and observational studies have suggested that dietary patterns such as the Mediterranean diet may alleviate OAB symptoms due to their anti-inflammatory and antioxidant properties (*Bozkurt et al., 2022*). Similarly, increased intake of dietary flavonoids (*e.g.*, anthocyanins, flavones) and sufficient vitamin D levels have been associated with improved urinary symptoms in clinical populations (*Chen et al., 2024a*; *Lin, Lyu & Feng, 2024*). These findings suggest that dietary factors may play a significant role in OAB pathophysiology. Fructose, a common dietary sugar, has recently garnered attention in OAB research. Fructose consumption has been implicated in metabolic syndrome, which shares pathophysiological features with OAB, including insulin resistance and chronic inflammation (*Hsu et al., 2022*). In rodent models, long-term high-fructose feeding (*e.g.*, 12 weeks) has been shown to induce insulin resistance and detrusor dysfunction, contributing to OAB-like symptoms (*Lee et al., 2021*). However, the precise mechanism linking fructose to OAB remains controversial. Experimental findings suggest several possible mechanisms, including impaired detrusor muscle contractility (*Lee et al., 2023*). In contrast, other research on bladder function in rats indicates that fructose consumption alone may not be the direct cause of OAB but instead exacerbates underlying metabolic disturbances, such as obesity and dyslipidemia, which may secondarily impair bladder function (*Wu et al., 2023*). Furthermore, an alternative hypothesis proposes that fructose intake may

contribute to OAB through chronic bladder ischemia and oxidative stress, independent of insulin resistance or obesity (*Lee et al., 2025*). These conflicting findings highlight the complexity of fructose's role in OAB and emphasize the need to clarify whether fructose directly contributes to OAB pathogenesis or exerts its effects indirectly through metabolic stress.

Existing OAB animal models include those based on metabolic syndrome, insulin resistance, induction, bladder outlet obstruction, cyclophosphamide-induced cystitis, hormone deficiency (*e.g.*, ovariectomy), and chronic stress exposure (*Chen et al., 2024b*; *Jung et al., 2018*; *Kitta et al., 2020*; *Ko et al., 2018*; *Lee et al., 2023*; *Leiria et al., 2013*; *Lu et al., 2023*; *Shen et al., 2021*; *Velasquez Flores et al., 2018*; *Wang et al., 2017*). However, these models primarily focus on adult populations or metabolic syndrome-associated mechanisms, with limited studies exploring the specific role of fructose consumption in pediatric OAB. Moreover, it remains unclear whether short-term high fructose intake can independently induce OAB symptoms in the absence of metabolic syndrome. Recent clinical research conducted by our team demonstrated a significant positive correlation between average daily fructose intake and the severity of OAB symptoms in children (*Cui et al., 2025*). In this study, we aimed to investigate whether short-term high fructose exposure in juvenile Sprague-Dawley (SD) rats could induce bladder dysfunction with OAB-like features, and to preliminarily explore associated molecular changes. Rather than validating a formal model, this exploratory study provides an initial framework to evaluate the bladder-related effects of high fructose feeding in early life. This approach aims to elucidate how fructose intake may contribute to bladder dysfunction and metabolic changes underlying OAB-like symptoms, offering a theoretical basis for future dietary intervention strategies.

## MATERIALS & METHODS

### Ethical statement and study approval

This study was approved by the Ethics Committee for Experimental Animals, Fujian Maternity and Child Health Hospital (IACUC-FMCHH-2024-016). All experiments were conducted in strict accordance with the guidelines and regulations for the use of laboratory animals set by the ethics committee. Furthermore, all procedures adhered to the ARRIVE (Animal Research: Reporting of *In Vivo* Experiments) guidelines 2.0 (*Perciedu Sert et al., 2020*).

### Animal modeling

Female Sprague-Dawley (SD) rats (Crl: CD(SD); RRID: RGD_734476), 3 weeks old, weighing 50–60 g (Sibefu Biotechnology Co., Ltd., Beijing, China), were weaned, separated from their dams, and housed for a 7-day acclimation period in a barrier environment to ensure their health and environmental adaptation. During this period, all rats were fed a purified maintenance diet (XTCON50B, XTM06-006, Jiangsu Xietong Pharmaceutical Bio-Engineering Co, Ltd., Nanjing, China) and had free access to water. The animals were housed in Polypropylene R3 cage (Zeya, Suzhou, Jiangsu, China; dimensions: 46.00 cm $\times$30.00 cm $\times$18.00 cm), with 15 rats per cage, under controlled environmental conditions

(temperature: 24 ± 0.5 °C; 12 h light/dark cycle), and provided with shredded paper bedding. Rats were monitored daily for signs of distress or injury, and any adverse events were recorded and reported. After the acclimation period, the rats were randomly assigned to two groups using a random number table: the model group ($n = 15$), which was fed 60% high-fructose purified diet (XT704, Jiangsu Xietong Pharmaceutical Bio-Engineering Co, Ltd., Nanjing, China), and the control group ($n = 15$), which were continued on the purified maintenance growth diet (XTCON50B). The nutritional composition of both diets is listed in Table 1. The 60% fructose formulation was selected based on previous studies to induce OAB-like symptoms in a short period (*Lee et al., 2023*). Both groups were were fed their respective diets for one week. At the end of the adaptation period, body weight was measured. 200 µL of blood was collected from the retro-orbital sinus under isoflurane anesthesia for analysis of glucose, triglycerides, cholesterol, insulin, fructose, and fructose-1-phosphate (F1P) levels. To assess whether short-term high fructose exposure could induce OAB-like symptoms, the following pre-specified phenotypic criteria were applied: (1) increased urinary frequency, as measured by void spot number in copper(II) sulfate ($CuSO_4 \cdot 5H_2O$) filter paper assays; (2) presence of unstable bladder contractions during the filling phase in urodynamic tests (*Hutchinson et al., 2020*); (3) reduced maximum bladder infusion volume, indicating compromised storage capacity; and (4) increased locomotor activity in the open field test. These criteria are consistent with commonly observed features in OAB models. However, we emphasize that the present study does not aim to establish a validated pediatric OAB model, but rather to determine whether short-term fructose feeding induces a constellation of symptoms resembling OAB and to explore accompanying molecular changes.

## Study design

After one week of dietary intervention, body weight was recorded, and open-field behavioral tests were conducted to assess the activity level and movement range of the rats. The open-field test is widely used to evaluate general activity and anxiety-related behavior. In the context of OAB-like symptoms, increased movement range, activity time, and total distance traveled are expected. Urine spots and voiding volumes were collected using metabolic cages. The number of void spots was recorded to evaluate urination frequency. Since rats tend to urinate in fixed locations, a higher number of void spots indicates increased voiding frequency and may reflect OAB-like symptoms.

Five rats from each group were randomly selected using a random number table to undergo oral glucose tolerance testing (OGTT) before and after the one-week dietary intervention, to determine whether short-term high-fructose exposure induced glucose intolerance in juvenile rats. Following OGTT, urodynamic studies were performed and analyzed using the BL-420N biological signal acquisition system (Chengdu Taimeng Biotechnology Co., Ltd., Sichuan, China). Additionally, six rats from the model group were randomly selected (random number table) for a Compound C intervention experiment. Dorsomorphin (Compound C, MCE, HY-13418A, Shanghai, China), an AMPK signaling pathway inhibitor, was administered intravenously at a dose of 10 µmol during the urodynamic test. No vehicle control was included. Rats exhibiting unstable contraction

**Table 1 Nutritional composition and ingredients of diets for control (XTCON50B) and model (XT704) groups.**

| Nutrition level | XTCON50B (Control) | | XT704 (Model) | |
|---|---|---|---|---|
| | gm% | kcal% | gm% | kcal% |
| Protein | 19.20 | 20.00 | 19.20 | 20.00 |
| Carbohydrate | 67.30 | 70.00 | 67.30 | 70.00 |
| Fat | 4.30 | 10.00 | 4.30 | 10.00 |
| Total | – | 100.00 | – | 100.00 |
| Energy (kcal/gm) | 3.85 | – | 3.85 | – |
| **Ingredient composition** | **gm** | **kcal** | **gm** | **kcal** |
| Casein | 200.00 | 800.00 | 200.00 | 800.00 |
| L-Cystine | 3.00 | 12.00 | 3.00 | 12.00 |
| Corn starch | 315.00 | 1,260.00 | 94.00 | 376.00 |
| Maltodextrin | 35.00 | 140.00 | 0.00 | 0.00 |
| Sucrose | 354.00 | 1,416.00 | 0.00 | 0.00 |
| Fructose | 0.00 | 0.00 | 610.00 | 2,440.00 |
| Cellulose | 50.00 | 0.00 | 50.00 | 0.00 |
| Soybean oil | 25.00 | 225.00 | 25.00 | 225.00 |
| Lard | 20.00 | 180.00 | 20.00 | 180.00 |
| Mineral mix (S10026B) | 50.00 | 0.00 | 50.00 | 0.00 |
| Vitamin mix (V10001C) | 1.00 | 4.00 | 1.00 | 4.00 |
| Choline bitartrate | 2.00 | 0.00 | 2.00 | 0.00 |
| FD&C Yellow Dye #5 | 0.05 | 0.00 | 0.00 | 0.00 |
| FD&C Blue Dye #1 | 0.00 | 0.00 | 0.05 | 0.00 |
| Total | 1,055.05 | 4,037.00 | 1,055.05 | 4,037.00 |

Notes.
XTCON50B, Purified maintenance diet; XT704, 60% high-fructose purified diet.

waves in two consecutive recordings were considered successfully modeled (*Xue et al., 2024*). At the end of the experiment, all rats were deeply anesthetized with isoflurane and then euthanized by cervical dislocation, in accordance with ethical guidelines. Euthanasia was confirmed by the absence of corneal reflex and cessation of respiration (*Abdelkhalek et al., 2021*). No rats were euthanized prior to the planned end of the experiment. Bladder were collected, weighed, and preserved for subsequent analyses.

## Urinary frequency measurement

Urinary frequency was assessed using a modified void spot assay (*Hill et al., 2018*) in metabolic cages (Braintree Scientific, Braintree, MA, USA) lined with copper(II) sulfate ($CuSO_4 \cdot 5H_2O$)-treated filter paper. The filter paper (10422-1005, Blicks) was soaked in saturated 2% $CuSO_4 \cdot 5H_2O$ solution (Sinopharm, 10003218) for 10 min and dehydrated at 200 °C for 1 h, following a previously described method (*Gu et al., 2014*). When contacted by urine, the dehydrated $CuSO_4 \cdot 5H_2O$ rehydrated and produced visible blue spots. During the 5-hour recording period (09:00–14:00, 25 ± 1 °C, 40%–60% relative humidity), rats had free access to water *via* a non-leaking, push-type sipper nozzle (ZH-DWZ-02, Zhenghua Lab Animal) to prevent water contamination of the filter paper (*Chen et al., 2017*). After
the test, the paper was photographed under ambient light, and urine spots were quantified using Photoshop (Adobe, reverse color contrast mode). Total urine output was estimated by weighing the filter paper before and after testing. Water intake during this test and feed intake per 24 h were recorded concurrently.

## Open field experiment

The open-field test was conducted to evaluate spontaneous locomotor activity, which may reflect behavioral hyperactivity associated with OAB-like symptoms. Previous studies have shown that changes in voiding behavior may be accompanied by increased motor activity, especially under conditions of bladder discomfort or urgency-like sensations (*Chen et al., 2022*). Therefore, the open-field test was used in this study to assess whether high-fructose intake affects overall activity levels in juvenile rats.

The open field test was conducted using an experimental system developed by Beijing Zhongshi Dichuang Technology Development Co., Ltd. Before starting the experiment, the testing chamber was thoroughly cleaned to ensure the removal of any odor or residue, including feces and urine from previous test subjects. Relevant parameters were set using the software, and basic information such as the animal identification number, experiment date, and status indicator data was recorded. The test was conducted between 15:00 and 17:00 in a quiet and temperature-controlled environment (22–24 °C).

During the experiment, the test animal was gently removed from its cage while ensuring the handler faced away from the animal to minimize stress. The animal was swiftly placed in the central area of the testing chamber, and the handler immediately exited. Behavioral analysis software automatically recorded the animal's activity duration and range within the chamber, including total distance of activity (cm), time of activity (s) and average speed (cm/s). The experiment lasted for 8 min. At the end of the experiment, the animal was returned to its prepared cage. The testing chamber was disinfected using an alcohol spray to eliminate residual odors, followed by wiping it dry with paper towels.

## OGTT

After an 8 h fasting period, the rats were orally administered glucose at a dose of one g/kg body weight. Blood samples were collected from the tail vein at 0 (baseline), 30, 60, 90, 120, and 180 min after glucose administration. Blood glucose levels at each time point were measured using a portable glucometer (Accu-Chek Performa, Roche Diagnostics, Indianapolis, IN, USA). Glucose tolerance was assessed by comparing blood glucose levels between the two groups at the respective time points.

## Urodynamic testing

Urodynamic testing was performed following a previously described method (*Fraser et al., 2020*). Rats were anesthetized using 2% isoflurane (maintained throughout the entire procedure *via* a nose cone inhalation system) gas and positioned supine upon achieving successful anesthesia. The surgical site was thoroughly disinfected with povidone-iodine. An F3 epidural catheter, pre-lubricated with paraffin oil, was gently inserted into the bladder of female rats *via* the urethra, as previously reported (*Fraser et al., 2020*). This transurethral catheterization approach is minimally invasive and does not require abdominal incision or

surgical exposure of the bladder. Residual urine in the bladder was aspirated by connecting the catheter to the bladder using a 5.0 ml syringe (MLT844, ADInstruments, Sydney, Australia), and pressure signals were continuously recorded using a BL420N physiological signal acquisition and analysis system (Chengdu Taimeng Software Co., Ltd., Sichuan, China). The catheter was then secured with adhesive tape and connected to a three-way device linking the epidural catheter, a microinjection pump, and a pressure measurement system. Urodynamic signals were recorded continuously throughout the entire bladder filling and post-drug administration period. All parameters were recorded in real time using LabChart software included the following: frequency of unstable contractions, intervals between filling contractions, amplitude of filling contractions (defined as the difference between peak and nadir pressures of contraction waves, measured in $cmH_2O$), maximum intravesical pressure during filling, maximum bladder filling volume, maximum pressure difference during filling (difference between initial filling pressure and maximum pressure), average pressure during filling, average pressure after filling cessation, pressure difference after filling cessation.

All pressure data were recorded in $cmH_2O$, time in seconds (s), and volume in milliliters (ml). The OAB-like features was confirmed if more than two unstable contractions (which were defined as transient intravesical pressure increases $> 5 \, cmH_2O$ during the filling phase without urine leakage) occurred in two consecutive filling experiments (*Abrams et al., 2002*). At this point, dorsomorphin was administered *via* tail vein at a dose of 10 μmol/kg body weight. Detailed urodynamic parameters were recorded post-injection. Finally, following ethical guidelines, rats were euthanized under deep isoflurane anesthesia and then euthanized by cervical dislocation. The bladder tissues were collected by dissection, immediately frozen in liquid nitrogen, and stored at −80 °C for biochemical analysis. Bladders used for biochemical analysis were selected from the urodynamics cohort to enable direct correlation between physiological bladder function and molecular endpoints. Similarly, blood was collected from the same animals for consistency in biochemical evaluation.

## Biochemical index measurement and animal samples

Following urodynamic experiments, blood samples were collected *via* retro-orbital puncture. After centrifugation at 1,510× g for 15 min, 200 μL of serum was collected for subsequent biochemical analyses, including cholesterol, triglycerides, glucose, insulin, fructose and F1P. The bladder was excised immediately, rinsed in cold saline, and dissected under a stereomicroscope to separate the mucosa and detrusor muscle using microforceps. Each tissue type was processed separately. Approximately 0.1 g of bladder tissue was homogenized in one mL of extraction buffer on ice. The homogenate was centrifuged at 13,720×g for 10 min at 4 °C, and the supernatant was collected for further analysis. Calcium ion concentrations in bladder tissues were measured using a calcium colorimetric assay kit (Elabscience, E-BC-K103-M, Wuhan, China). Fructose levels were quantified using the Fructose Content Detection Kit (Grace Biotechnology, G0530W, Suzhou, China). F1P levels were measured using a commercial ELISA kit (Shanghai Enzyme Linked Biotechnology, YJ266599, Shanghai, China). Serum levels of total cholesterol and

triglycerides were measured using ELISA kits (cholesterol: Nanjing Jiancheng, A111-1-1; TG: Nanjing Jiancheng, A110-1-1), and insulin levels were determined using an Insulin ELISA kit (Nanjing Jiancheng, MM-0587R2, Nanjing, China). All assays were performed strictly in accordance with the manufacturers' instructions.

## Hematoxylin and eosin staining

Bladder tissue sections were formalin-fixed and embedded in paraffin prior to sectioning. Sections were deparaffinized in xylene and rehydrated through a graded series of ethanol solutions. Nuclei were stained with hematoxylin for 10 min and rinsed under running tap water. Cytoplasm staining was performed by immersing the sections in 0.5% eosin solution for 30 s to 1 min. The sections were then dehydrated, cleared, and mounted with coverslips. All HE-stained sections were viewed under a light microscope (DM2500, Leica Microsystems) at 400× magnification. For each animal, five randomly selected non-overlapping fields were analyzed. Slide scoring was performed by an investigator blinded to group allocation.

## Immunohistochemistry

Paraffin-embedded bladder tissue sections (four μm thick) were immersed in citrate buffer (0.01 M, pH = 6.0) and subjected to heat-induced antigen retrieval at 95 °C for 5 min. Immunostaining was performed using primary antibodies against MYH-10 (ab230823, 1:2000; Abcam, Cambridge, UK), MYH-11 (ab224804, 1:500; Abcam, Cambridge, UK), and SMA (smooth muscle actin; ab108531, 1:100; Abcam, Cambridge, UK). A DAB (3,3′-diaminobenzidine) kit was used for chromogenic development, and nuclei were counterstained with hematoxylin. Images were captured using a DP70 camera. The staining intensity was scored on a four-level scale: 0 (negative, no staining), 1 (weak positive, light yellow), 2 (positive, yellowish-brown), and 3 (strong positive, brownish-brown). The percentage of positive cells was also scored: ≤ 25% (1 point), 26–50% (2 points), 51–75% (3 points), and >75% (four points). The final score was calculated by multiplying the staining intensity score by the percentage of positive cells. Image acquisition and scoring were conducted at 400× magnification. Five random non-overlapping fields per section were analyzed for each animal. All evaluations were performed by an investigator blinded to experimental group assignment.

## Immunofluorescence staining

Reactive oxygen species (ROS) in tissue sections were detected using a ROS detection kit (BB-470516, Shanghai Biobay Biomedical Engineering Co., Ltd.; Shanghai, China). Frozen sections were prepared using OCT embedding compound. The experimental procedures strictly followed the manufacturer's instructions. Tissue sections were placed in a humidified chamber, and 100–200 μL of ROS probe working solution was applied to each section. The sections were incubated at 37 °C in the dark for 40 min. Excess staining solution was removed, and the sections were washed 2–3 times with washing solution. At room temperature, 200 μL of washing solution was gently added to cover the entire tissue section. After 1–2 min of incubation, the excess washing solution was carefully removed, and the tissue sections were allowed to air dry completely. Finally, coverslips were

mounted using glycerol, and the sections were observed under a fluorescence microscope (DM6000B-FL, Leica Microsystems). Images were captured at 400× magnification, and five non-overlapping regions per section were analyzed per animal.

## Western blot

Bladder tissues were carefully dissected, and the mucosa and detrusor muscle were separated under a stereomicroscope. Homogenized bladder mucosa and detrusor muscle samples from each group were lysed in cell lysis buffer (P0013, Beyotime, Shanghai, China) supplemented with protease and phosphatase inhibitor mixture (P1045, Beyotime, Shanghai, China). Total protein concentrations were determined using the Pierce 660 nm Protein Assay. Proteins were separated by sodium dodecyl sulfate-polyacrylamide gel electrophoresis (SDS-PAGE) in Laemmli buffer system and transferred onto PVDF membranes (IPVH00010, Millipore, Burlington, MA, USA). Equal amounts of protein (30 μg) were loaded per lane for SDS-PAGE analysis. The membranes were probed with the following primary antibodies: TRPV1 (ab305299, 1:1000; Abcam, Cambridge, UK), CaMKK2 (ab96531, 1:500; Abcam, Cambridge, UK), phosphorylated AMPK (p-AMPK, ab109402, 1:1000; Abcam, Cambridge, UK), AMPK (ab32047, 1:1000; Abcam, Cambridge, UK), and β-actin (ab8226, one μg/ml; Abcam, Cambridge, UK). After washing, membranes were incubated with horseradish peroxidase (HRP)-conjugated secondary antibodies, including HRP Goat Anti-Rabbit IgG (H+L) (AS014; ABClonal, Woburn, MA, USA) and HRP Goat Anti-Mouse IgG (H+L) (AS003; ABClonal, Woburn, MA, USA), for 1 h at room temperature. Protein bands were detected using enhanced chemiluminescence (ECL) reagent (ACE Aisiyi Biological, AL010-01) and visualized on a ChemiDoc imaging system (Bio-Rad). All procedures were performed in accordance with the protocols provided by ABclonal Biology Co., Ltd. (Woburn, MA, USA).

## Energy metabolomics analysis of bladder muscle tissue

After thawing and grinding, 0.05 g of tissue samples were mixed with 500 μL of 70% methanol/water solution (1.06007.4008, Merck, Rahway, NJ, USA). The mixture was vortexed at 700×g for 3 min and centrifuged at 13,720×g at 4 °C for 10 min. A total of 300 μL of the supernatant was transferred to a new centrifuge tube and stored at −20 °C for 30 min. The sample was then centrifuged again at 13,720× g at 4 °C for 10 min. Following centrifugation, 200 μL of the supernatant was transferred through a protein precipitation plate and prepared for further liquid chromatography-mass spectrometry (LC-MS) analysis.

## Ultra-performance liquid chromatography and electrospray ionization mass spectrometry/mass spectrometry

The sample extracts were analyzed using an LC-electrospray ionization mass spectrometry/mass spectrometry (ESI-MS/MS) system, consisting of a Waters ACQUITY H-Class UPLC (https://www.waters.com/nextgen/us/en.html) coupled with a QTRAP® 6500+ mass spectrometer (https://sciex.com/). For UPLC analysis, the ACQUITY UPLC BEH Amide column (2.1 × 100 mm, 1.7 μ m) was employed. The mobile phase consisted of solvent A, composed of water containing 10 mM ammonium acetate (AA-6236-0250,

Sigma-Aldrich, St. Louis, MO, USA) and 0.3% ammonium hydroxide (A112080-500 ml, Sigma-Aldrich, St. Louis, MO, USA), and solvent B, consisting of 90% acetonitrile/water (V/V) (CAEQ-4-000308-4000, Merck). Formic acid (R050195-500 ml, Sigma-Aldrich, St. Louis, MO, USA) was used where applicable for pH adjustment and ionization efficiency. The gradient elution program was set as follows: 0−1.2 min, 95% B; 8 min, 70% B; 9–11 min, 50% B; 11.1–15 min, re-equilibration to 95% B. The flow rate was maintained at 0.4 mL/min, the column temperature was set to 40 °C, and the injection volume was two μL.

For mass spectrometry analysis, linear ion trap (LIT) and triple quadrupole (QQQ) scanning data were acquired using the QTRAP® 6500+ LC-MS/MS system equipped with an ESI Turbo IonSpray interface. Both positive and negative ion modes were utilized, and the system was controlled *via* Analyst 1.6.3 software (Sciex). The ESI source operating conditions were as follows: the ion source was ESI+/-; the source temperature was 550 °C; the spray voltage was 5,500 V in positive ion mode and −4500 V in negative ion mode; and the curtain gas (CUR) was set at 35 psi. Energy metabolism and associated metabolites were analyzed using scheduled multiple reaction monitoring (MRM), with specific MRM transitions monitored within designated time windows based on the retention times of individual metabolites. Data acquisition was conducted using Analyst 1.6.3 software, and quantification of metabolites was performed with Multiquant 3.0.3 software (Sciex). Declustering potential (DP) and collision energy (CE) for each MRM transition were optimized to enhance detection sensitivity. The final processed data provided detailed profiles of energy metabolism and its related compounds.

## Data analysis

The following statistical and bioinformatics analyses were performed exclusively for the metabolomics dataset. Unsupervised principal component analysis (PCA) was performed using the statistical function prcomp in R software (http://www.r-project.org), with data scaled to unit variance prior to PCA. Hierarchical clustering analysis (HCA) and Pearson correlation coefficient (PCC) analysis were conducted to explore the relationships between samples and metabolites. HCA results were visualized as heatmaps with dendrograms, while PCC values between samples were calculated using the cor function in R software and presented as heatmaps. Both HCA and PCC visualizations were generated using the heatmap package in R. For HCA, the normalized signal intensities of metabolites (scaled to unit variance) were represented using a color gradient. Metabolomics raw data are available in the File S1.

Significant differential metabolites between groups were identified based on variable importance in projection (VIP) values and absolute $\log_2$ fold change ($\log_2$FC). VIP values were extracted from orthogonal partial least squares discriminant analysis (OPLS-DA), which also included score plots and permutation tests. OPLS-DA analysis and associated visualizations, such as score plots and permutation plots, were performed using the R package MetaboAnalystR. Data were mean-centered prior to OPLS-DA, and to prevent overfitting, a 200-time permutation test was conducted. Kyoto Encyclopedia of Genes and Genomes (KEGG) annotation and enrichment analysis were performed to understand

the biological relevance of identified metabolites. Annotated metabolites were mapped to the KEGG compound database (http://www.kegg.jp/kegg/compound/) and subsequently to KEGG pathway databases (http://www.kegg.jp/kegg/pathway.html). Pathways with significantly regulated metabolites were subjected to metabolite set enrichment analysis, with significance determined by the $P$-value from a hypergeometric test.

## Statistical analysis

All statistical analyses were performed using SPSS version 23.0 software (IBM Corp., Armonk, NY, USA). Categorical variables were summarized as frequencies and percentages, whereas continuous variables were expressed as mean $\pm$ standard deviation (mean $\pm$ SD). Independent samples $t$-tests were used for comparisons between two groups. For repeated measures, two-way analysis of variance (two-way ANOVA) with time and group as factors was employed to evaluate the interaction effects between treatment and time. For visualization clarity, $P$ values at individual time points were displayed where significant differences were observed. Statistical significance was defined as $P < 0.05$, with significance levels denoted as * $P < 0.05$, ** $P < 0.01$, *** $P < 0.001$.

## RESULTS

### Short-term high fructose intake affects bladder metabolic indicators in juvenile SD rats

After 1 week of treatment, the body weight of the model rats ($148.70 \pm 10.48$) was significantly lower compared to the control group ($159.20 \pm 10.90$) ($P < 0.01$, Fig. 1A). While no significant difference in blood cholesterol was observed between the control group ($1.04 \pm 0.47$) and model group ($1.04 \pm 0.72$) after 1 week (Fig. 1B), blood triglycerides levels in the model group ($1.64 \pm 0.34$) was significantly reduced after 1 week compared to the control group ($2.36 \pm 0.86$) ($P < 0.001$, Fig. 1C), suggesting that high fructose intake may rapidly alter lipid metabolism in juvenile SD rats. Blood glucose and insulin levels showed no significant differences between the two groups. However, blood insulin levels in the model group ($5.25 \pm 4.44$) were significantly increased compared to their pre-experiment levels ($2.13 \pm 0.25$) ($P < 0.001$, Fig. 1D), while blood glucose levels in the model group ($5.73 \pm 0.80$) were significantly decreased compared to their pre-experiment levels ($6.38 \pm 0.75$) ($P < 0.05$, Fig. 1E). Blood fructose levels were significantly increased in both the control and model groups after one week ($P < 0.001$, Fig. 1F). However, there was no significant statistical difference in fructose levels between the two groups. The blood F1P levels showed no significant differences between the two groups or before and after the experiment (Fig. 1G). The blood glucose levels of each group were measured at various time points during the OGTT before and after one week. At 0 min, no significant differences were observed between the groups. At 60 min, significant differences were observed between the Mod-0w and Con-0w groups ($P < 0.05$, Fig. 1K). Although a statistically significant difference was observed at 120 min, the overall OGTT response after 1 week showed no consistent difference between the Mod-1w and Con-1w groups (Fig. 1L). This isolated difference at 120 min may reflect individual variability, as no consistent trend was observed

across time points. Overall, there were no significant differences in glucose tolerance between the 1-week fructose feeding group and the control group in the OGTT.

Intracellular calcium ion levels in the bladder muscle tissue of the model group ($0.37 \pm 0.25$) were significantly increased compared to the control group ($0.10 \pm 0.04$) after 1 week ($P < 0.05$, Fig. 1H), which may contribute to altered bladder excitability after fructose exposure. However, fructose content and F1P levels in bladder muscle tissue showed no significant differences between the two groups (Figs. 1I and 1J).

## Short-term high fructose intake affects urinary and behavioral patterns in juvenile SD rats

The number of urine imprint points was significantly higher in the model group ($11.20 \pm 3.12$) than in the control group ($5.13 \pm 2.10$; $P < 0.001$, Figs. 2A–2B). Water intake over 5 h was also significantly increased in the model group ($6.74 \pm 1.87$ mL) compared to controls ($5.28 \pm 1.38$ mL; $P < 0.05$, Fig. 2C). In contrast, no significant differences were observed between groups in 24-hour feed intake or total urine volume following urinary frequency assessment (Figs. 2D–2E). These findings suggest that both groups had comparable access to food and similar feeding behavior under identical environmental conditions. In the open field test, the total distance of activity, time of activity, and average speed were significantly higher in the model group compared to the control group ($P < 0.001$, Figs. 2F, 2G and 2H).

## Short-term high fructose intake induces bladder dysfunction in juvenile SD rats

Compared to the control group, bladder weight, bladder capacity, and residual intravesical pressure after voiding showed no significant differences in the model group (Figs. 3A–3C). However, the interval between non-voiding contractions during the filling phase was significantly shortened (Fig. 3D), and the amplitude of these contractions was markedly increased (Fig. 3E), indicating enhanced involuntary bladder activity. Additionally, the maximum infusion volume was significantly reduced in the model group (Fig. 3F), suggesting impaired bladder storage function. Urodynamic traces revealed more frequent non-voiding contractions during the filling phase in the model group, which were partially alleviated by Compound C treatment (Fig. 3G). Although maximum and average bladder pressures during the filling phase were not significantly altered (Figs. 3H–3I), the amplitude of cessation of peristaltic contraction was significantly increased in the model group (Fig. 3J), implying abnormal contractile termination during voiding reflexes.

In the intervention study, treatment with Compound C significantly prolonged the interval between non-voiding contractions in the model group (Fig. 4A), suggesting a reduction in bladder overactivity. No significant differences were observed in the amplitude of non-voiding contractions, maximum infusion volume, maximum or average bladder pressures, or the amplitude of cessation of peristaltic contraction between the two groups (Figs. 4B–4F), indicating that Compound C primarily modulated afferent excitability during bladder filling without affecting other cystometric parameters.

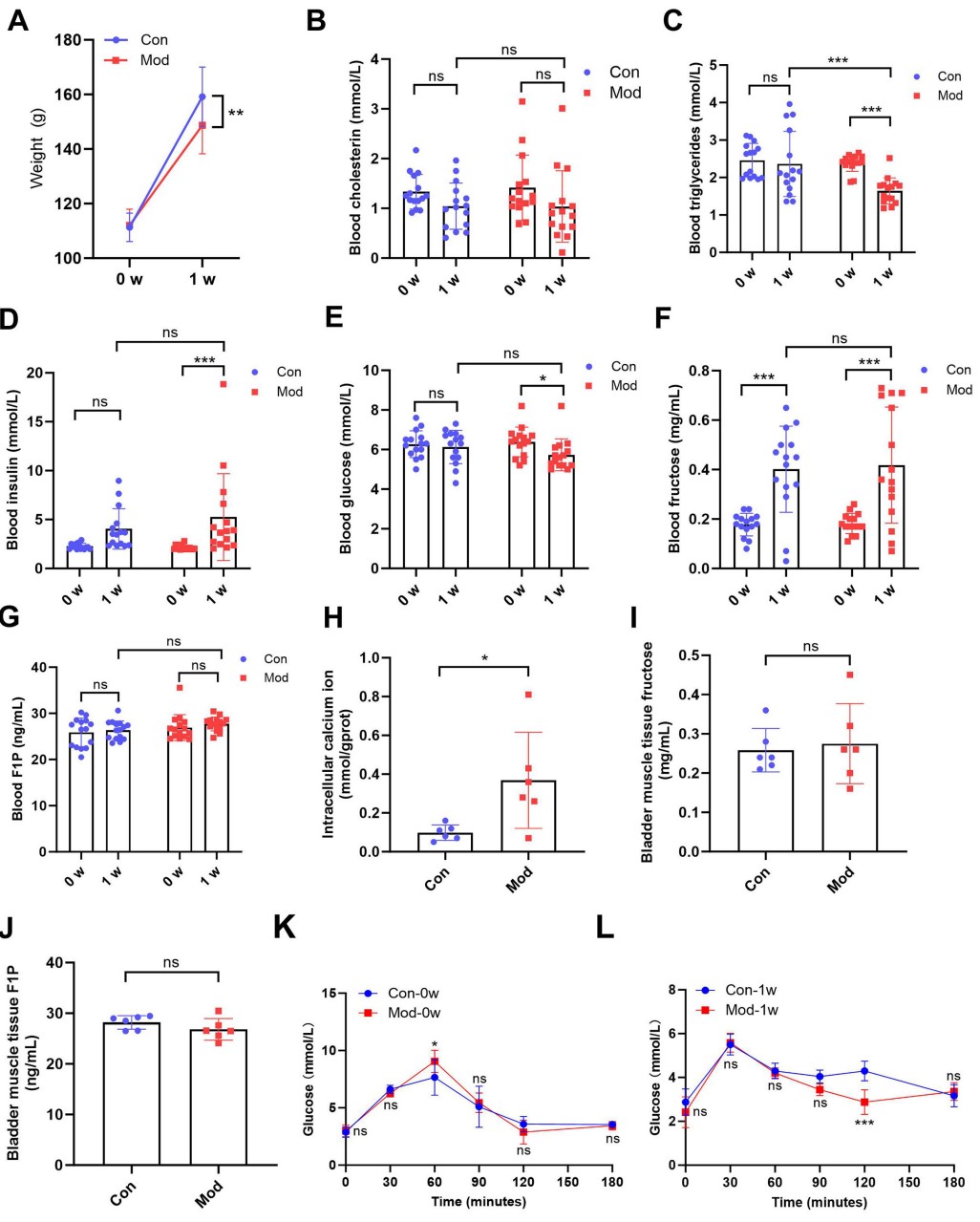

**Figure 1  Short-term high fructose intake affects bladder metabolic indicators in juvenile SD rats.**
(A) Changes in body weight of juvenile SD rats ($n = 15$); (B) Changes in blood cholesterol ($n = 15$); (C) Changes in blood triglycerides levels ($n = 15$); (D) Changes in blood insulin levels ($n = 15$); (E) Changes in blood glucose levels ($n = 15$); (F) Changes in blood fructose levels ($n = 15$); (G) Changes in blood F1P levels ($n = 15$); (H) Intracellular calcium ion content in bladder muscle tissue ($n = 6$); (I) Fructose content in bladder muscle tissue ($n = 6$); (J) F1P content in bladder muscle tissue ($n = 6$). (K) Blood glucose levels during the oral glucose tolerance test (OGTT) before fructose feeding (0w) ($n = 6$); (L) Blood glucose levels during the OGTT after one week of fructose feeding (1w) ($n = 6$). Abbreviations: Con, control group; Mod, model group; F1P, fructose-1-phosphate. * $P < 0.05$, ** $P < 0.01$, *** $P < 0.001$ compared to the Con group; ns indicates no statistical difference.

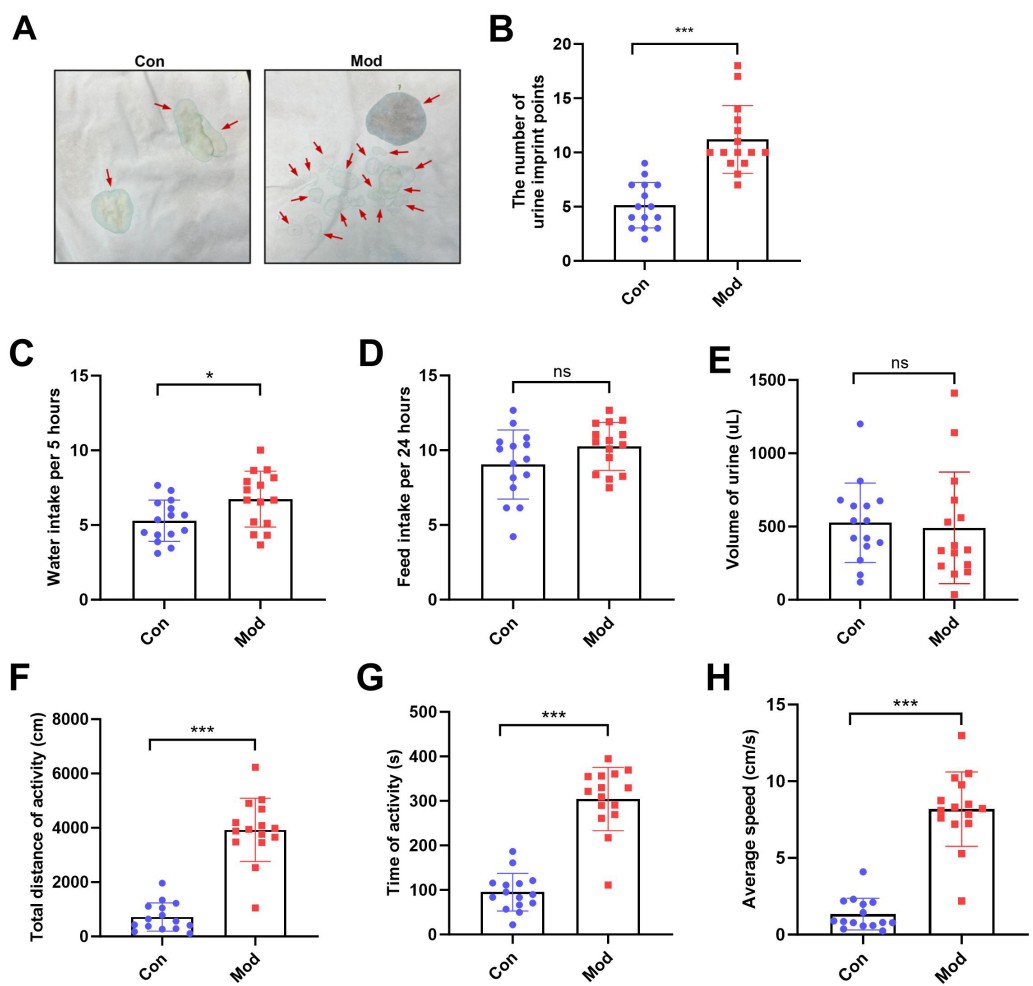

**Figure 2  Short-term high fructose intake affects urinary and behavioral patterns in juvenile SD rats.**
(A) Urinary frequency measurement using the copper (II) sulfate ($CuSO_4 \cdot 5H_2O$) filter paper assays ($n = 15$); (B) The number of voiding events recorded over 5 h using a metabolic cage ($n = 15$); (C) Water intake measured over 5 h using a metabolic cage ($n = 15$); (D) Food intake measured over 24 h ($n = 15$); (E) Volume of urine over 5 h using a metabolic cage ($n = 15$); (F) Open field test results showing total distance of activity ($n = 15$), (G) time of activity ($n = 15$), (H) average speed ($n = 15$). Abbreviations: Con, control group; Mod, model group. * $P < 0.05$, ** $P < 0.01$, *** $P < 0.001$ compared to the Con group; ns indicates no statistical difference.

## Short-term high fructose intake induces bladder smooth muscle abnormalities and increased oxidative stress

HE staining revealed that the bladder tissues of the model group exhibited disorganized smooth muscle cell arrangement, disrupted structural integrity between cells, increased interstitial matrix, and vacuolar-like changes (Fig. 5A). Immunohistochemical staining for SMA showed enhanced staining in the model group, indicating increased activity and an expanded distribution of smooth muscle cells, which suggests elevated metabolic activity of smooth muscle cells (Fig. 5B). MYH-10 staining showed significantly increased expression in the model group, with enhanced staining observed in both interstitial cells

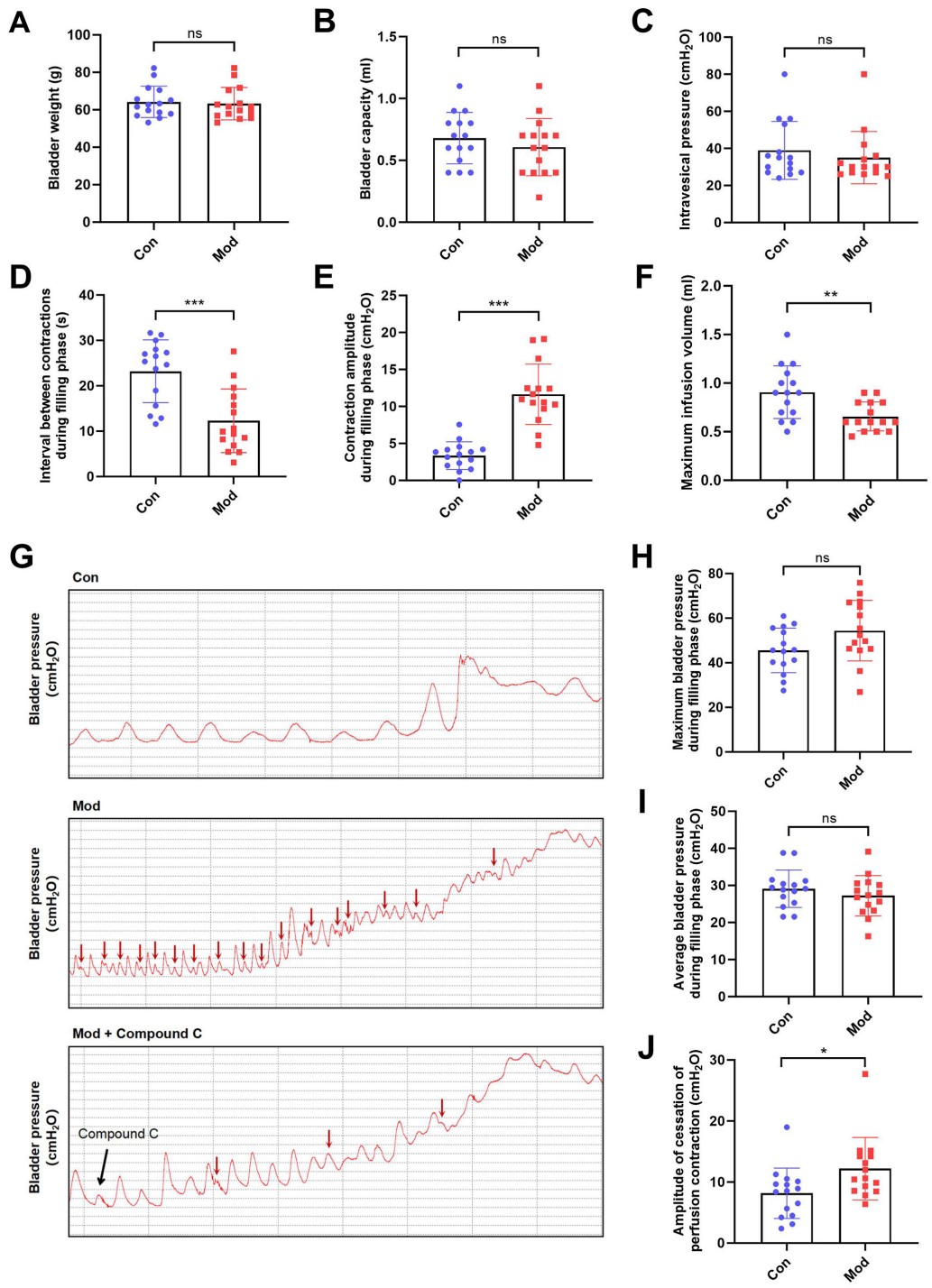

**Figure 3** **Short-term high fructose intake induces bladder dysfunction in juvenile SD rats between control and model group.** (A) Bladder weight ($n = 15$); (B) Bladder capacity, represented by intravesical pressure at the start of voiding ($n = 15$); (C) Residual intravesical pressure after voiding ($n = 15$); (D) Interval between non-voiding contractions during the filling phase (continued on next page...)

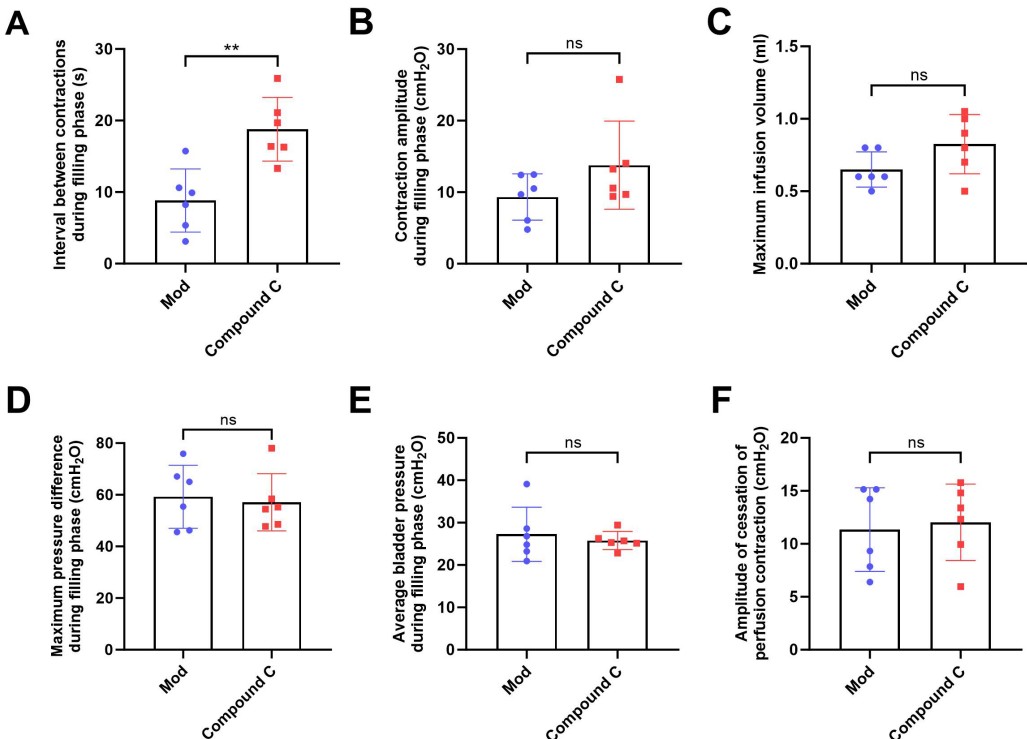

**Figure 3 (...continued)**
($n = 15$); (E) Amplitude of non-voiding contractions during the filling phase ($n = 15$); (F) Maximum infusion volume during cystometry ($n = 15$); (G) Bladder pressure fluctuations recorded during urodynamic testing (the red arrows indicate non-voiding contractions) in control ($n = 15$), model ($n = 15$), and model plus Compound C groups ($n = 6$); (H) Maximum bladder pressure during the filling phase ($n = 15$); (I) Average bladder pressure during the filling phase ($n = 15$); (J) Amplitude of cessation of peristaltic contraction ($n = 15$). Abbreviations: Con, control group; Mod, model group. * $P < 0.05$, ** $P < 0.01$, *** $P < 0.001$ compared to the Con group; ns indicates no statistical difference.

**Figure 4** **Short-term high fructose intake induces bladder dysfunction in juvenile SD rats between model and Compound C group.** (A) Interval between non-voiding contractions during the filling phase ($n = 6$); (B) Amplitude of non-voiding contractions during the filling phase ($n = 6$); (C) Maximum infusion volume during cystometry ($n = 6$); (D) Maximum bladder pressure during the filling phase ($n = 6$); (E) Average bladder pressure during the filling phase ($n = 6$); (F) Amplitude of cessation of peristaltic contraction ($n = 6$). Abbreviations: Mod, model group. ** $P < 0.01$ compared to the Con group; ns indicates no statistical difference.

and smooth muscle cells, indicating increased activity of interstitial cells and potential tissue remodeling (Fig. 5C). In contrast, smooth muscle myosin heavy chain (SMMHC) staining was significantly reduced in the model group, indicating decreased expression of contractile proteins, which suggests that the contractile function of smooth muscle cells may be impaired or abnormal (Fig. 5D). MitoSOX Red fluorescence staining demonstrated significantly elevated mitochondrial ROS levels in the model group, indicating increased oxidative stress (Fig. 5E).

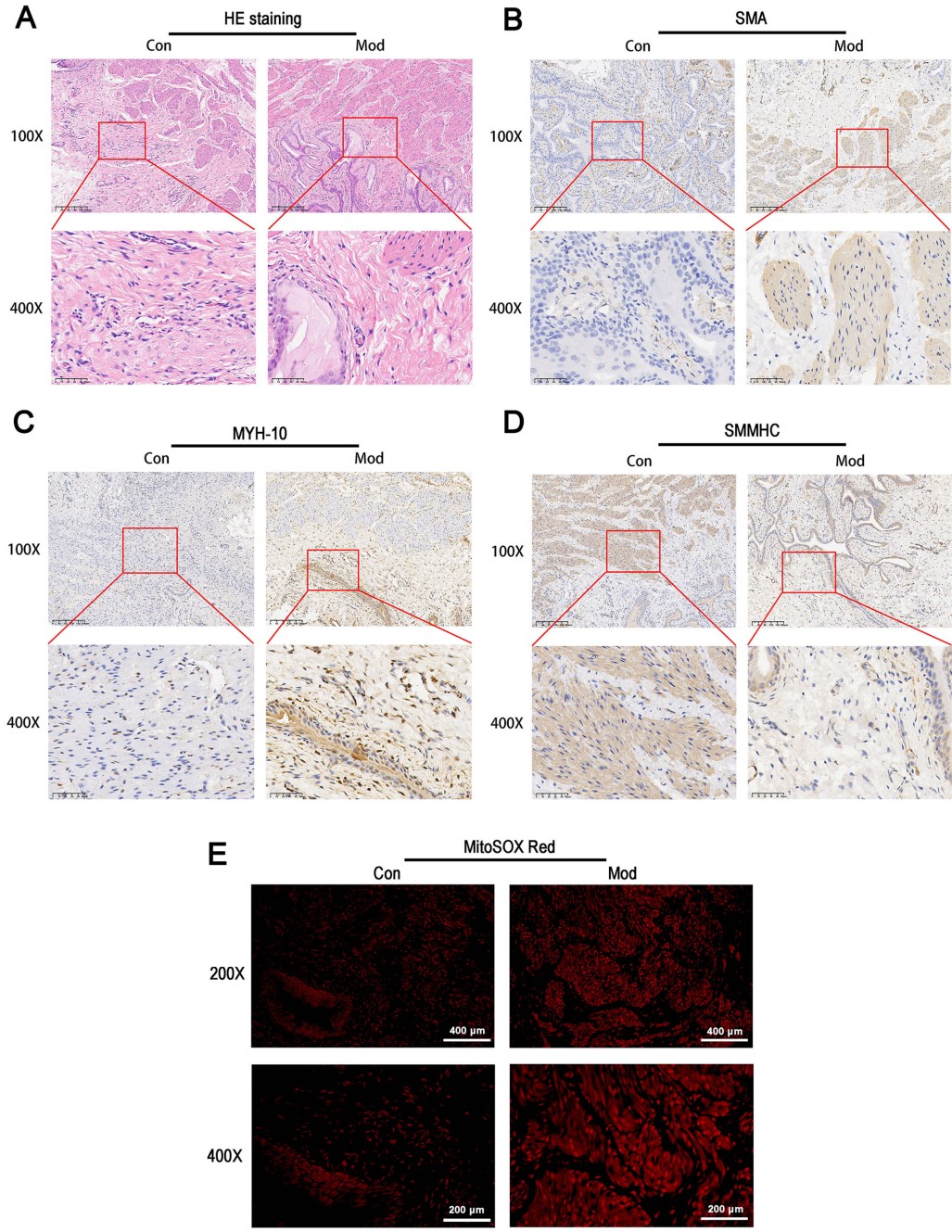

**Figure 5 Short-term high fructose intake induces bladder smooth muscle abnormalities and increased oxidative stress.** (A) Structural changes in smooth muscle and interstitial cells in bladder tissues observed *via* HE staining (scale bar = 200 μm; enlarged view scale bar = 50 μm); (B) Distribution and marker expression of smooth muscle cells detected using SMA immunohistochemical staining (scale bar = 200 μm; enlarged view scale bar = 50 μm); (C) Distribution and expression of non-muscle cells detected *via* MYH-10 immunohistochemical staining (scale bar = 200 μm; enlarged view scale bar = 50 μm); (D) Expression of contractile proteins in smooth muscle cells detected using SMMHC immunohistochemical staining (scale bar = 200 μm; enlarged view scale bar = 50 μm); (continued on next page…)

**Figure 5 (…continued)**
(E) Mitochondrial ROS levels in bladder tissues detected *via* MitoSOX Red fluorescence staining (scale bar = 400 μm; enlarged view scale bar = 200 μm). Abbreviations: HE, hematoxylin-eosin; SMA, smooth muscle actin; MYH-10, myosin heavy chain-10; SMMHC, smooth muscle myosin heavy chain; MitoSOX Red, mitochondrial ROS fluorescence probe.

## Short-term high fructose intake significantly alters the metabolic profile of bladder smooth muscle tissue in juvenile SD rats

PCA demonstrated a clear separation between the model group and the control group along the PC1 and PC2 axes, indicating significant differences in metabolite profiles between the two groups (Fig. 6A). A heatmap of metabolites revealed distinct relative abundances between groups, with many metabolites showing significant changes in the Mod group (Fig. 6B). $Log_2FC$ bar plots identified significantly altered metabolites, with key metabolites in the Mod group exhibiting markedly higher fold changes compared to the Con group (Fig. 6C). Radar plot analysis further highlighted metabolic network changes in the Mod group, revealing significant upregulation of critical metabolites such as 2,3-Diphosphoglyceric acid and ATP within metabolic pathways (Fig. 6D).

Gene ontology (GO) classification further demonstrated that metabolites in the Mod group were predominantly associated with metabolic processes, particularly those involved in energy production and redox regulation (Fig. 7). KEGG pathway analysis revealed that the metabolites in the Mod group were significantly enriched in carbon metabolism, glycolysis/gluconeogenesis, the tricarboxylic acid (TCA) cycle, and the pentose phosphate pathway, all of which play critical roles in energy metabolism and redox balance (Fig. 8).

Specifically, ATP levels in the Mod group were significantly higher than those in the Con group (Mod: 5,747.16 ng/ml, Con: 612.3 ng/ml, $Log_2FC$: infinity, $P < 0.001$), while ADP levels were significantly lower in the Mod group (Mod: 0 ng/ml, Con: 1,303.43 ng/ml, $Log_2FC$: negative infinity, $P = 0.088$), indicating a high-energy state in the bladder muscle cells (Fig. 6B).

According to *Aimaretti et al. (2023)*, different muscle tissues exhibit varying adaptive responses to high-fructose diets, and bladder dysfunction may be associated with mitochondrial dysfunction caused by oxidative stress. Interestingly, our Mod group samples showed significant upregulation of glycolysis-related metabolites, indicating enhanced intracellular glycolysis. However, the pyruvate-to-phosphoenolpyruvate (PEP) ratio was significantly decreased (Mod: 0.085, Con: 0.165), accompanied by a marked increase in lactate production.

Metabolites within the TCA cycle, such as malate and citrate, were significantly elevated, although the increase in succinate was less pronounced. Recent studies suggest that lactate exposure can lead to increased ROS generation and upregulation of genes associated with calcium ($Ca^{2+}$) signaling, including those linked to CaN and $Ca^{2+}$/calmodulin-dependent kinase (CaMK). ROS is known to elevate intracellular $Ca^{2+}$ levels and enhance CaMK activity. In our study, lactate levels in the Mod group were significantly increased (Mod: 107,677.22 ng/ml, Con: 61,202.82 ng/ml; $Log_2FC$: 0.82; $P = 0.0042$). However, the lactate-to-pyruvate ratio showed a decreasing trend (Mod: 56.11, Con: 64.8), indicating both increased accumulation and consumption of lactate in the Mod group.

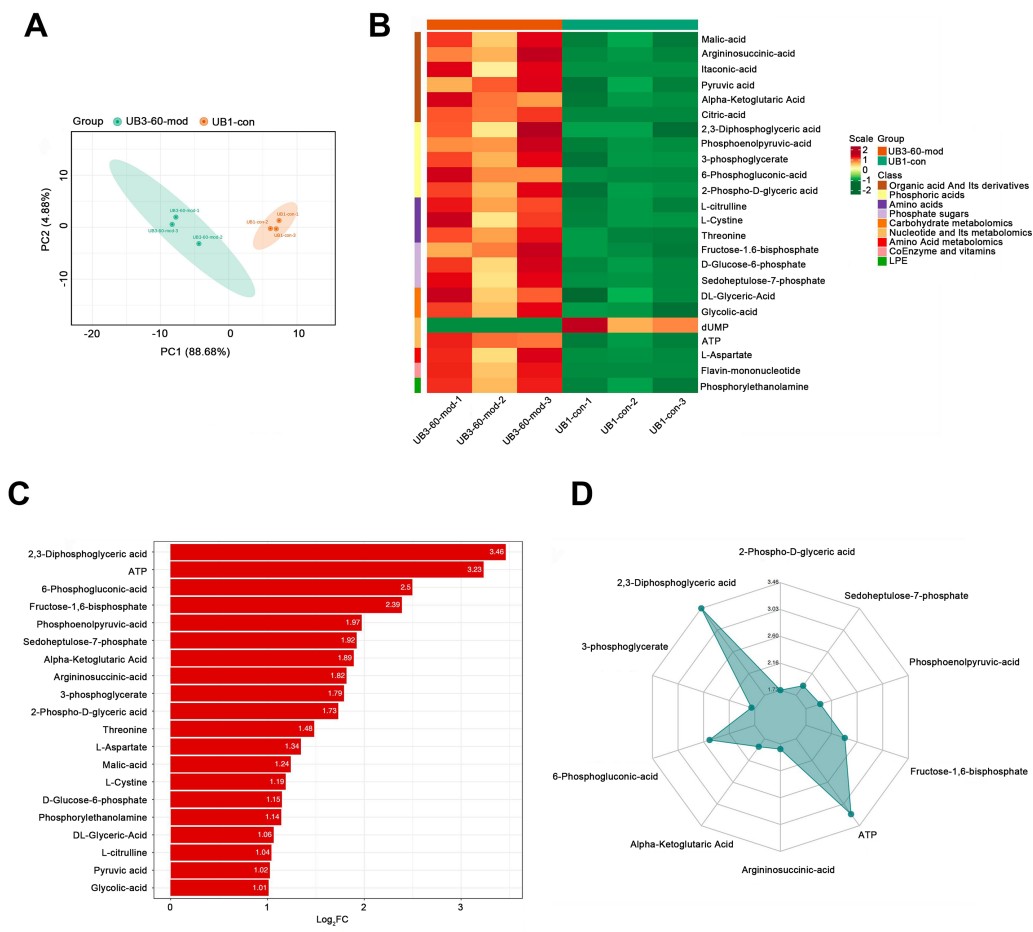

**Figure 6  Short-term high fructose intake significantly alters the metabolic profile of bladder smooth muscle tissue in juvenile SD rats.** (A) Principal component analysis (PCA) of bladder smooth muscle metabolites from the control (Con) and model (Mod) groups, demonstrating the degree of clustering and separation of metabolite distributions between groups; (B) Heatmap showing the relative abundance distribution of differential metabolites across groups, with categorized trends for each metabolite; (C) Log₂ Fold Change (Log₂FC) bar plot displaying fold changes of differential metabolites, highlighting those with significant differences; (D) Radar plot illustrating metabolite changes in the Mod group within the metabolic network, revealing impacts on key metabolic pathways. Abbreviations: PCA, Principal Component Analysis; Log₂FC, logarithm (base 2) of fold change.

This suggests that the cells exhibited enhanced oxidative stress, resulting in excessive ROS production. In the metabolic network, activation of the pentose phosphate pathway was also evident. Levels of metabolites such as 6-phosphogluconate, D-ribulose-5-phosphate, D-xylulose-5-phosphate, and sedoheptulose-7-phosphate were significantly elevated, suggesting an increased glycolytic flux. This also points to enhanced amino acid biosynthesis, nucleic acid metabolism, and genetic information transfer capacity within cells. Moreover, the PPP plays a critical role in mitigating oxidative stress by generating NADPH, a protective factor that alleviates oxidative damage induced by high fructose intake.

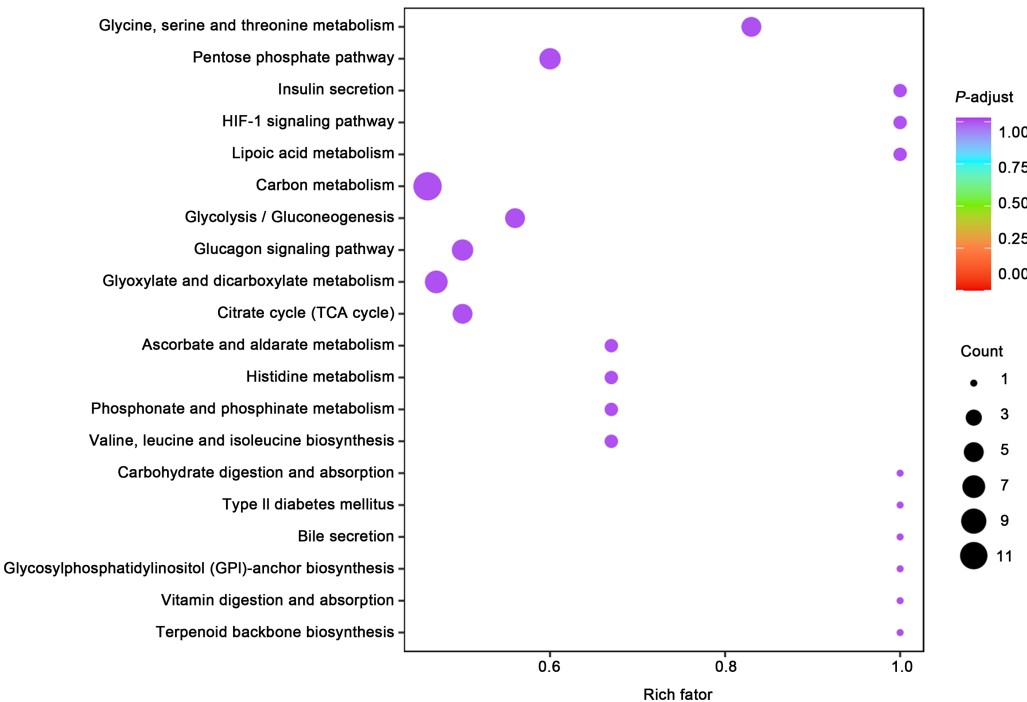

**Figure 7** **GO analyses reveal functional enrichment of metabolites in bladder smooth muscle tissue of juvenile SD rats.** Gene Ontology (GO) classification displaying the functional enrichment of metabolites in the Mod group, predominantly in metabolic pathways related to energy production and redox processes. The distribution of categorized metabolites is visualized by colors and percentages across different functional categories.

## Short-term high fructose intake reshapes the metabolic network in bladder muscle tissue of juvenile SD rats

Heatmap analysis revealed distinct distribution patterns of metabolites in bladder smooth muscle tissue between the Con and Mod groups. Key metabolites involved in carbohydrate metabolism, phosphorylated acids, and their derivatives, such as glycerol-3-phosphate and ATP, were significantly upregulated in the Mod group (Fig. 9A). Among metabolites related to amino acid metabolism, arginine, glutamine, and serine were markedly elevated in the Mod group compared to the Con group (Fig. 9B). Analysis of nucleotide metabolism and related metabolites indicated significant upregulation of metabolites such as hypoxanthine, phosphoethanolamine, and the coenzyme FMN in the Mod group, highlighting critical metabolic nodes affected by short-term high-fructose intake (Fig. 9C).

## Short-term high fructose intake activates the CaMKK2/AMPK signaling pathway in bladder smooth muscle tissue

The above results suggest that in the model group of rats, under the action of fructose, there were manifestations of enhanced catabolism and inhibited lipidogenesis. These results indicate that AMPK is likely to be phosphorylated and activated under short - term high - fructose diet.

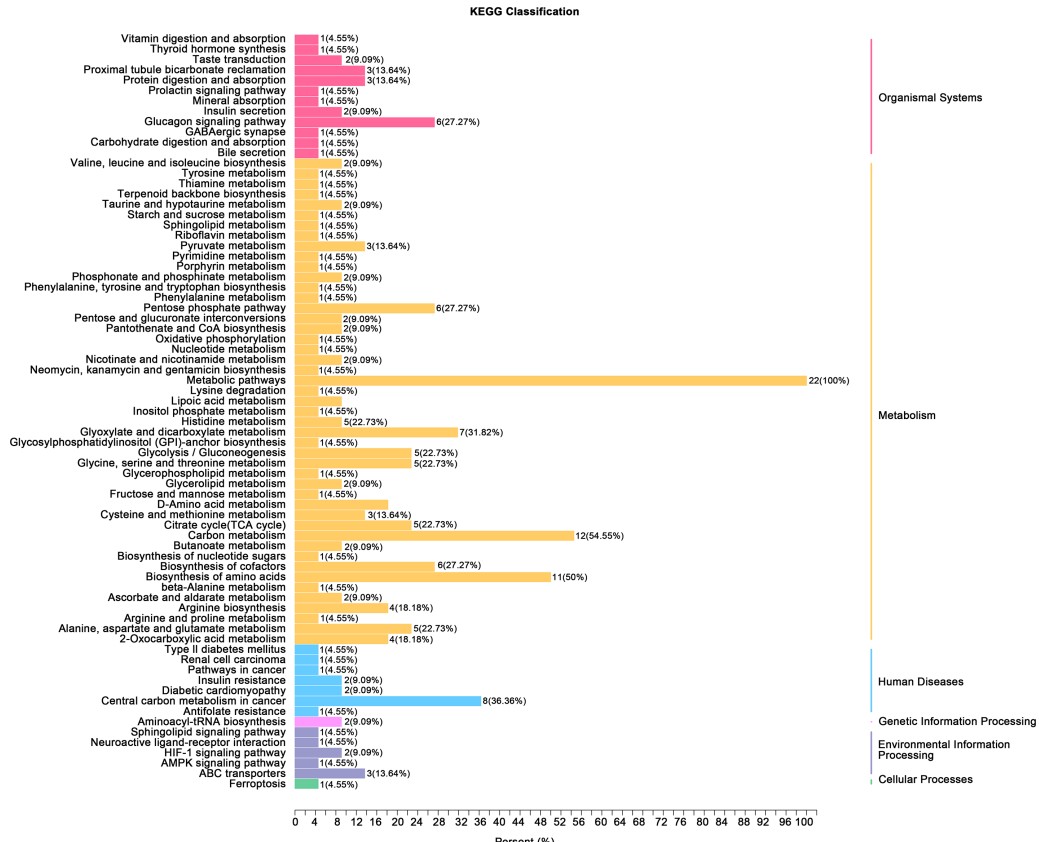

**Figure 8  KEGG analyses reveal enriched metabolic pathways in bladder smooth muscle tissue of juvenile SD rats.** Kyoto Encyclopedia of Genes and Genomes (KEGG) analysis evaluating the enrichment of Mod group metabolites in significant metabolic pathways, including carbon metabolism, glycolysis/gluconeogenesis, the pentose phosphate pathway, and the tricarboxylic acid (TCA) cycle. The color of the dots represents the *P*-value, and the dot size indicates the number of metabolites involved.

The protein expression of the CaMKK2/AMPK signaling pathway was analyzed by Western blot. The results showed that TRPV1 protein expression in the model group was not significantly different compared to the control group. CaMKK2 protein expression in the model group showed a significant increase compared to the control group ($P < 0.001$, Fig. 10). The ratio of p-AMPK to AMPK in the model group also showed a significant increase compared to the control group ($P < 0.001$, Fig. 10).

## DISCUSSION

Currently, no universally accepted animal model accurately reproduces the pathophysiological mechanisms of pediatric OAB induced by high fructose intake. Existing models primarily focuson metabolic syndrome and are based on adult rodents. For example, *Velasquez Flores et al. (2018)* demonstrated that excessive succinate intake leads to OAB symptoms in a metabolic syndrome model. Similarly, *Lee et al. (2016)* reported that maternal and early-life fructose consumption leads to metabolic syndrome and OAB-like

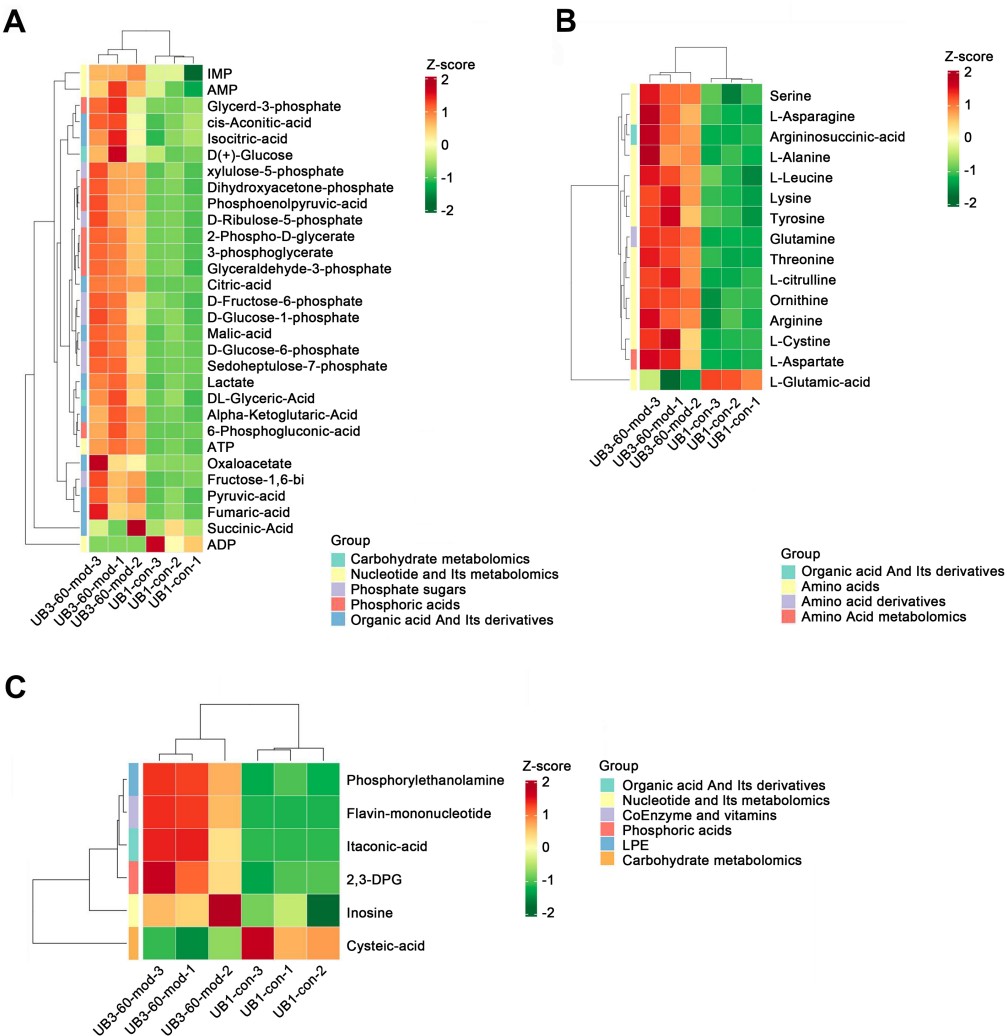

**Figure 9** **Short-term high fructose intake reshapes the metabolic network in bladder muscle tissue of juvenile SD rats, significantly affecting amino acid, carbohydrate, and nucleotide metabolism.** (A) Heatmap showing distribution patterns of metabolites in bladder smooth muscle tissue between the Con and Mod groups, highlighting changes in key metabolites involved in carbohydrate metabolism, phosphorylated acids, and their derivatives; (B) Heatmap of amino acid-related metabolites, displaying differences in relative abundance and distribution between the Con and Mod groups, with a focus on amino acids and their derivatives; (C) Heatmap analysis of nucleotides, coenzymes, and vitamins, illustrating significant expression changes in key metabolites in the Mod group. Abbreviations: LPE, lysophosphatidylethanolamine; Z-score, standard score.

symptoms in adult offspring, with a strong association between fructose intake and insulin resistance. However, these studies rely on long-term fructose feeding (up to 13 weeks), which does not reflect the rapid symptom onset seen in pediatric OAB.

*Gatto et al. (2023)* reported that even short-term high fructose intake causes severe mitochondrial dysfunction in the skeletal muscles of juvenile rodents. Based on this, we investigate whether one-week high fructose consumption induces OAB-like symptoms in

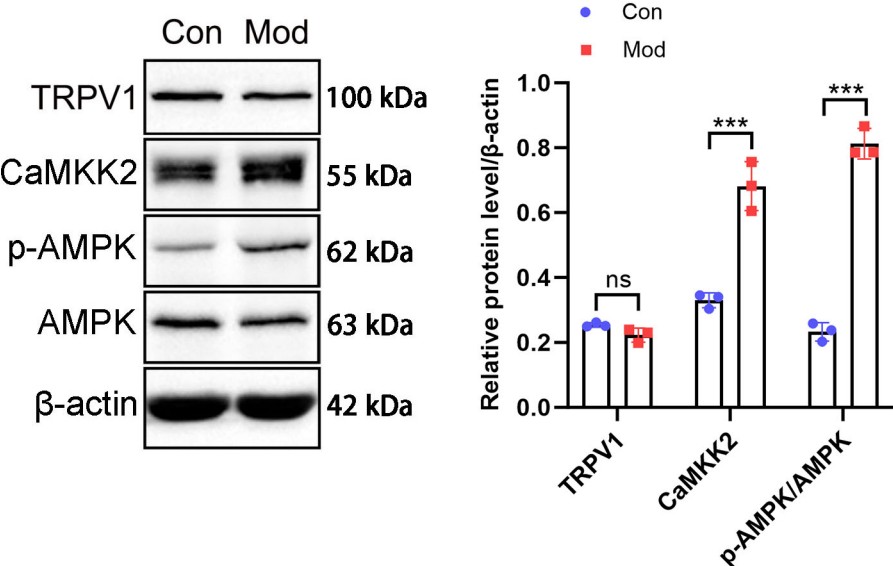

**Figure 10** **Short-term high fructose intake induces bladder smooth muscle abnormalities and increased oxidative stress.** Western blot was used to detect the protein expression of TRPV1, CaMKK2, p-AMPK, and AMPK, with β-actin used as an internal control. Abbreviations: Con, control group; Mod, model group. *** $P < 0.001$ compared to the Con group; ns indicates no statistical difference.

juvenile SD rats. While fructose feeding is not a classical method for modeling OAB, our aim was to evaluate whether short-term exposure could induce bladder dysfunction with OAB-like features rather than to formally validate a standard model. After one week, no significant differences in blood glucose or insulin levels were observed between the model and control groups, suggesting that OAB symptoms may occur independently of insulin resistance or metabolic syndrome. During the 5-hour urinary frequency measurement period, animals had free access to water, and water intake was recorded. Previous studies have shown that water availability does not significantly influence voiding behavior in rodents (*Chen et al., 2017*; *Wegner et al., 2018*). Although a difference in water intake was observed between groups, no significant difference in total urine volume was found. These findings suggest that the increased number of voids observed in the model group is unlikely to be attributable to more water consumption alone, but rather reflects altered voiding patterns or bladder function. Urodynamic tests revealed frequent unstable bladder contractions during the filling phase in the model group, while detrusor activity remained stable in control group. These results indicate that OAB-like symptoms were successfully induced within a short timeframe. However, combining phenotypic characterization with mechanistic exploration within a single study may limit interpretability. Thus, mechanistic insights presented here should be viewed as preliminary and require further validation. Voiding spot assays revealed that control rats typically voided in spatially confined areas, producing fewer, concentrated spots, whereas the model group showed a significantly greater number of spatially dispersed spots despite similar urine volumes. This distribution pattern reflects altered bladder function rather than fluid-driven diuresis. In open field

tests, increased locomotor speed and activity range were consistent with urodynamic findings, providing additional behavioral evidence of OAB-like symptoms (*Soeda et al., 2021*). Histological and molecular analyses revealed structural alterations in the bladder, including interstitial cell proliferation and cytoplasmic changes in the muscle layer. The expression of smooth muscle marker MYH11 and interstitial cell marker MYH10 indicated a distinct imbalance in cell populations, differing from the characteristics of neurogenic bladder models (*Weyne et al., 2018*). However, the specific effects of fructose on cellular function and secretion require further investigation.

Fructose toxicity, characterized by oxidative stress and mitochondrial dysfunction, is closely associated with cellular damage in various tissues (*Fang et al., 2021*). Studies have shown that excessive fructose metabolism leads to the accumulation of F1P, glyceraldehyde, and other metabolites, triggering the production of ROS (*Xu et al., 2022*). In this study, although F1P levels in bladder muscle tissue did not significantly increase, downstream metabolites such as GA and dihydroxyacetone phosphate were markedly elevated, indicating a "fructose escape" phenomenon, where unmetabolized fructose spills into peripheral tissues. The significant accumulation of ROS and the reduced pyruvate/PEP ratio observed in the model group further suggest support the involvement of oxidative stress and hypoxia as key contributors to OAB-like symptoms. The sharp rise in 2,3-diphosphoglycerate levels indicates heightened oxygen consumption, implying potential hypoxic stress. This observation is consistent with findings by *Taylor et al. (2021)*, who demonstrated that under hypoxic conditions, fructose-exposed cells exhibit elevated F1P levels, decreased pyruvate/PEP ratios, and inhibited activity of the hypoxia-adaptation enzyme PKM2, thereby enhanced HIF-1α activation and improved cellular survival in hypoxic environments. Similar metabolic alterations were observed in the model group, including significant lactate accumulation, which may act as an additional source of ROS. Metabolic analysis revealed activation of glycolysis, the TCA cycle, the pentose phosphate pathway, and amino acid metabolism in bladder muscle tissue. *Li et al. (2023)* reported that in cardiomyocytes deficient in Cpt1b, the utilization of long-chain fatty acids was effectively inhibited, leading to a shift toward alternative energy production pathways. Likewise, *Liu et al. (2020)* employed uniformly labeled $^{13}$C-glucose and $^{13}$C-fructose in primary hepatocytes to reveal that fructose not only contributes to glycolysis by providing substrates for energy production through the TCA cycle but also serves as a carbon source for nucleotide synthesis (*e.g.*, sedoheptulose-7-phosphate (S7P) and inosine monophosphate (IMP)) and the production of various amino acids. In this study, metabolites related to amino acid metabolism, particularly those involved in the ammonia and urea cycles, were significantly elevated in the model group, indicating upregulated activity in these pathways. Rats in the model group exhibited reduced body weight and lower serum triglyceride levels, suggesting enhanced fatty acid oxidation and diminished anabolic metabolism. Interestingly, serum fructose also increased in the control group, likely due to endogenous metabolic changes and the breakdown of dietary sucrose (354 g) in purified maintenance diet. Taken together, these findings suggest that short-term high-fructose intake promotes tissue catabolism and induces bladder dysfunction through fructose escape and localized fructose-driven metabolic stress.

In addition to its classical activation by cellular energy deficiency, AMPK in this study appears to be primarily activated by oxidative stress and intracellular calcium accumulation. AMPK serves as a central regulator of cellular energy homeostasis, restoring ATP levels by promoting catabolic pathways and inhibiting anabolic processes during energy scarcity (*Dengler, 2020*). Oxidative stress increases intracellular $Ca^{2+}$ levels, which activates CaMKKβ and subsequently triggers AMPK activation, representing a pathway more closely linked to redox imbalance than to conventional energy depletion (*Yang et al., 2024*; *Zhu et al., 2022a*). In partial bladder outlet obstruction (PBOO) models, AMPK activation has been linked to detrusor instability. For example, astragaloside IV (AS-IV) has been shown to protect detrusor muscle from PBOO-induced oxidative damage by activating mitophagy through the AMPK-ULK1 pathway, thereby improving contractility and reversing bladder wall remodeling (*Zhu et al., 2022b*). Moreover, the CaMKKβ/AMPK signaling axis plays a pivotal role in modulating detrusor contractile function. In PBOO-induced bladder hypertrophy, both CaMKKβ expression and AMPK phosphorylation are significantly reduced, contributing to impaired bladder contractility (*Choi et al., 2019*). Pharmacological inhibition of CaMKKβ or AMPK in healthy bladders has been shown to increase detrusor contractility, further underscoring this pathway's regulatory significance (*Choi et al., 2019*). Dorsomorphin has been used in bladder cancer studies to inhibit AMPK. In T24 and EJ cells, it enhanced oleanolic acid-induced cytotoxicity by blocking AMPK-dependent autophagy (*Song et al., 2017*). In FXR-overexpressing urothelial carcinoma cells, it reversed AMPK-mediated suppression of migration and angiogenesis (*Lai et al., 2022*). These findings support the relevance of AMPK inhibition in bladder-derived cells. Although earlier studies suggest that AMPK activation can mitigate bladder overactivity in ischemia and cystitis models (*Yang et al., 2021b*; *Zhang et al., 2016*), our findings indicate that under high-fructose conditions, AMPK activation, likely triggered by oxidative stress and elevated intracellular $Ca^{2+}$, contributes to detrusor instability. This apparent contradiction may reflect differences in the underlying pathological mechanisms, disease duration, or model context. Unlike ischemic models where AMPK is suppressed, high-fructose feeding results in its metabolic activation, potentially shifting AMPK's role from protective to maladaptive. These findings highlight the context-dependent nature of AMPK signaling in bladder physiology (*Yang et al., 2021a*). AMPK inhibition by Dorsomorphin significantly prolonged the interval between non-voiding contractions, suggesting a reduction in bladder afferent excitability. Other cystometric parameters such as contraction amplitude, bladder pressure, and infusion volume remained unaffected, indicating a selective effect on non-voiding activity rather than global changes in detrusor function. While our study primarily focuses on the impact of high fructose intake on AMPK activation, previous research has shown that AMPK is associated with bladder contraction function in the PBOO model, providing further context for understanding the role of AMPK in bladder function. In this study, the activation of AMPK is primarily associated with intracellular calcium ion accumulation and oxidative stress, rather than through the conventional energy deficiency signaling pathway. MitoSOX Red fluorescence staining revealed a significant increase in mitochondrial ROS levels, indicating that oxidative stress plays a central role in the activation of the AMPK pathway following high fructose intake. Additionally, the

antagonism of AMPK significantly altered bladder contraction rhythms, reduced unstable contractions, and enhanced contraction amplitude, highlighting the important role of AMPK in regulating bladder function.

TRPV1, a transient receptor potential channel primarily expressed in the bladder mucosa, affecting detrusor overactivity (*Zhao et al., 2021*). In the bladder, TRPV1 is involved in sensing mechanical or chemical stimuli, such as tension during bladder distention or local oxidative stress (*Du et al., 2024*). Although no significant changes in TRPV1 expression were observed in bladder smooth muscle, its potential role in calcium signaling and metabolism in other tissues should be explored further. Studies have indicated that TRPV1 activation may influence AMPK activation through calcium signaling pathways, and this mechanism could play a role in the metabolic changes induced by a high fructose diet. TRPV1, known for its role in nociception and heat sensation, also impacts intracellular calcium signaling and glucose metabolism (*Scherer et al., 2017*). For instance, TRPV1 activation has been associated with increased glucose oxidation and ATP production in skeletal muscle cells *via* CaMKK2 and AMPK activation (*Vahidi Ferdowsi et al., 2021*). Additionally, eugenol-mediated activation of TRPV1 in a type II diabetes mellitus model increased intracellular calcium levels and activated CaMKK2 and AMPK, enhancing glucose uptake and metabolic adaptation (*Jiang et al., 2022*). Our findings support previous studies showing that high fructose intake activates the CaMKK2/AMPK pathway, though no significant changes in TRPV1 expression were observed in bladder smooth muscle. Given that AMPK is a central regulator of cellular energy balance, its activation in bladder smooth muscle may represent an adaptive response to metabolic stress induced by excessive fructose intake (*Muraleedharan & Dasgupta, 2022*). This is further supported by the known role of AMPK in enhancing mitochondrial function (*Wu & Zou, 2020*), reducing oxidative stress (*Li et al., 2024*), and promoting energy-efficient metabolic pathways (*Steinberg & Hardie, 2023*), which may help mitigate fructose-induced dysfunction in detrusor muscle.

In this study, short-term high-fructose feeding in juvenile Sprague-Dawley rats resulted in metabolic and functional alterations of the bladder consistent with OAB-like features. However, several limitations should be noted. First, the duration of fructose exposure was relatively short, and no experimental verification of symptom reversibility following fructose withdrawal was conducted. Second, although fasting insulin levels were measured before and after fructose feeding, insulin sensitivity was not assessed using an insulin tolerance test or HOMA-IR analysis. This limits our ability to determine the systemic metabolic impact of fructose exposure. Future studies incorporating these tests will be important for evaluating the potential contribution of insulin resistance to bladder dysfunction. Third, while fructose was found to modulate metabolites associated with oxidative stress and AMPK pathway activation in bladder tissue, further mechanistic studies are needed. These include investigating the accelerated catabolism of F1P, mapping the distribution of fructose-derived carbon in bladder tissue, identifying key metabolic enzymes responsible for intermediate accumulation, and clarifying the role of AMPK activation in detrusor contractility. Additionally, the increase in interstitial cell populations within bladder muscle observed in this study raises new questions about cell and cell interactions, particularly how these changes affect smooth muscle function. Fourth, as an exploratory

study designed to generate hypotheses, no corrections for multiple comparisons were applied. This limits the strength of statistical inference, as some observed differences may result from random variation. Given the small sample size and number of variables analyzed, all *p*-values should be interpreted descriptively and the findings considered preliminary (*Amrhein, Greenland & McShane, 2019*). Lastly, although the model reproduced several features consistent with pediatric OAB, its translational relevance to human disease remains to be confirmed. Future studies should include clinical validation to determine whether similar metabolic and molecular mechanisms are present in children with OAB and to assess the therapeutic potential of targeting fructose metabolism and AMPK signaling in clinical settings. Additionally, although we measured fasting insulin levels before and after fructose exposure, insulin sensitivity was not evaluated *via* an insulin tolerance test, which limits our understanding of how fructose feeding may have affected systemic insulin responsiveness. Future studies incorporating insulin tolerance test or HOMA-IR analysis will be essential to comprehensively assess insulin resistance and its potential role in bladder dysfunction.

## CONCLUSIONS

This study demonstrated that short-term high-fructose intake in juvenile SD rats induced bladder dysfunction and detrusor instability, presenting features consistent with OAB-like symptoms. These changes were associated with oxidative stress, calcium ion accumulation, and activation of the CaMKK2/AMPK signaling pathway, suggesting a potential mechanistic link between fructose metabolism and bladder dysfunction. Further research is needed to confirm the causal relationships among these pathways. The findings provide a basis for exploring metabolic contributions to pediatric OAB-like symptoms and support future development of targeted interventions focusing on metabolic regulation and oxidative stress mitigation.

## ACKNOWLEDGEMENTS

We would like to extend our sincere gratitude to Professor Caiwen Duan and Professor Liang Zheng from Shanghai Children's Medical Center for their invaluable guidance and assistance in project establishment as well as experimental implementation. The technology support provided by Ai Spiral Biotechnology Co., Ltd. in Fujian, China is greatly appreciated. The authors used AI-assisted tools (ChatGPT) to support grammar correction and linguistic clarity improvement. All scientific content was written and verified by the authors.

### Funding

This research was supported by the Fujian Provincial Natural Science Foundation (No. 2024J011124); the Provincial Medical Innovation Double High-Level Project (No. ETK2022003 and ETK2023007); and the Fujian Science and Technology Innovation Joint

Fund Project (No. 2024Y9555). The funders had no role in study design, data collection and analysis, decision to publish, or preparation of the manuscript.

## Grant Disclosures
The following grant information was disclosed by the authors:

The Fujian Provincial Natural Science Foundation: No. 2024J011124.

The Provincial Medical Innovation Double High-Level Project: No. ETK2022003, ETK2023007.

The Fujian Science and Technology Innovation Joint Fund Project: No. 2024Y9555.

## Competing Interests
The authors declare there are no competing interests.

## Author Contributions
- Xu Cui conceived and designed the experiments, performed the experiments, analyzed the data, prepared figures and/or tables, authored or reviewed drafts of the article, and approved the final draft.
- Zhiqiang Chen performed the experiments, analyzed the data, prepared figures and/or tables, and approved the final draft.
- Longyao Xu performed the experiments, analyzed the data, prepared figures and/or tables, and approved the final draft.
- Changwei Wu performed the experiments, analyzed the data, prepared figures and/or tables, and approved the final draft.
- Chaoming Zhou conceived and designed the experiments, performed the experiments, analyzed the data, prepared figures and/or tables, authored or reviewed drafts of the article, and approved the final draft.

## Animal Ethics
The following information was supplied relating to ethical approvals (*i.e.*, approving body and any reference numbers):

The Ethics Committee for Experimental Animals of Fujian Maternity and Child Health Hospital provided full approval (IACUC-FMCHH-2024-016) for this research.

## Data Availability
The metabolomics raw data are available in the Supplementary File.

## Supplemental Information
Supplemental information for this article can be found online at http://dx.doi.org/10.7717/peerj.20186#supplemental-information.

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
