# Peer review of "Metabolic profile alterations in juvenile rats with bladder overactivity induced by short-term high-fructose intake"

_PeerJ, doi:10.7717/peerj.20186_

## Round 0.1 · original submission · Major Revisions

Two experts had evaluated this manuscript. Both found it to be a good topic. However, they also raised significant concerns and suggestions for further improving it.

**Language Note:** The review process has identified that the English language must be improved. PeerJ can provide language editing services - please contact us at [email protected] for pricing (be sure to provide your manuscript number and title). Alternatively, you should make your own arrangements to improve the language quality and provide details in your response letter. – PeerJ Staff


Reviewer 1 ·

Basic reporting

Please check the text for academic writing accuracy.

Experimental design

The article discusses the development of an overactive bladder model using fructose supplementation in young rats. The study presents several evaluations aimed at understanding how the model behaves. Below are some suggestions that may improve the methodology and ensure the model’s reproducibility by other research groups.

Validity of the findings

To enhance the reproducibility of the model by other research groups—given that the article describes the development of an animal model of overactive bladder using a 60% fructose diet for a short period in young rats—it is necessary to include additional information in the text to improve methodological transparency and replicability.

Additional comments

Materials and methods:
Line 96: The text can be rewritten as: "Female Sprague-Dawley (SD) rats, 3 weeks old, weighing 50–60 g." This structure is clear and standardized, ensuring that the age and characteristics of the animals are well defined without ambiguities.
Line 98: What is the appropriate cage size to accommodate 15 rats? Specify the dimensions of the cages where the animals were housed.
Line 95: The Animal and experimental design section can be divided for better understanding. Additionally, the description of experimental procedures can be moved to a separate section starting from line 111 to improve the organization of the text and facilitate the reader’s comprehension.
Line 104: What is the nutritional composition of the diet used in the study? Presenting this information in a table would be useful. What is the rationale for using 60% fructose? Was the energy intake calculated?
Line 131 - Urinary frequency measurement:
• How was the filter paper used in the experiment obtained? Was it purchased or specifically produced for the study?
• Is there a reference for the type of filter paper used? This information is crucial to ensure reproducibility by other research groups.
• Does only urine leave blue marks on the paper?
• Did the animals have access to water during the test? If so, how were possible water spills distinguished from urine volume?
• What were the environmental conditions in the experiment room (temperature, humidity)?
• At what time of day were the tests conducted?
• Were the animals pre-conditioned to the metabolic cage?
• Was the filter paper placed before or after urine collection?
• During the 10-hour experiment period, did the animals have access to water? It is necessary to clarify whether the filter paper and urine collection occurred in the presence or absence of water.
These details must be included in the text to allow replication by other research groups.
Line 131: Was water intake monitored during the urinary behavior assessment or only during the treatment phase?
Line 138 - Open Field Experiment:
• Specify the time at which the test was conducted.
• Indicate which parameters were evaluated in the method section.
Line 153: How was the OGTT calculated? Describe the methodology used for the test.
• Why was the insulin tolerance test not performed?
Line 157 - Urodynamics:
• Were the animals kept under isoflurane anesthesia throughout the experiment? This information is essential for interpreting the results.
• Specify whether the urodynamics test was conducted with anesthetized or awake animals.
• Justify the choice of catheter placement in the urethra rather than at the bladder apex.
• Was the abdomen sutured or left open?
• What pressure measurement and data acquisition system was used?
• Were urodynamics recorded continuously or just a single measurement?
• Why were the bladders used in the urodynamics test selected for biochemical analyses? Was the effect of anesthesia on bladder function considered?
Line 178:
• What was the rationale for selecting animals from the urodynamics test for biochemical analysis of blood and bladder samples?
• Was the effect of anesthesia on these parameters considered?
Line 208:
• Specify whether the tissue sections were prepared using paraffin or another fixation method (paraffin, OCT, etc).
Line 220:
• The phrase "Homogenized bladder mucosa and detrusor muscle samples from each group were lysed" does not clarify whether the mucosa was separated from the detrusor muscle prior to extraction or if the entire bladder was processed as a whole.
• This section of the text should be revised to explicitly state whether a separation method was used.
Results and Discussion
1. How do the authors explain the increase in fructose levels in the control group between days 0 and 7?
2. Could the increase in the number of voids observed on the filter paper be related to significant water intake prior to the experiment?
3. Additionally, how were non-voiding contractions defined? It is necessary to specify in the text the minimum pressure in cmH₂O considered for these contractions.
4. Were six rats used for the Dorsomorphin evaluation after collecting urodynamic data from the 15 animals? Why wasn't this compound administered to the control group as well?
5. At what age were the rats weaned in this study? Was the fructose-enriched diet initiated immediately after weaning?
6. Why was the use of Dorsomorphin not discussed in the text?
7. Why was the number of voids not recorded during the urodynamic evaluation? Was the examination not continuous?
8. Why wasn’t the average urine volume of each spot on the filter paper calculated? This could be related to bladder capacity observed during urodynamic.
Figures and Tables
• Why are the urodynamic data split between Tables 3 and 4? Presenting the data in a single, unified table would facilitate interpretation.
• Table titles show missing or illegible characters—it is important to review formatting to ensure clear understanding.
• The experimental group number should be included in each figure legend.

Reviewer 2 ·

Basic reporting

1. Figure 1: On day 0, both groups should be the same. However, for one time point in the OGTT, a low p-value is reported. I have no general problem with that; it just shows that groups can differ based on chance alone if many comparisons are made. However, the OGTT time course after 1 week of fructose also differs at only one time point. Why would one be a chance finding, whereas the other can be interpreted as a biological alteration?

2. The figures would be easier to read if the indicator of the central tendency (horizontal line) were thicker. Moreover, an SD error bar to that line would also be helpful. I found the panels in Figure 2 to be more helpful. The continuous data in Figure 7 should also be presented as in Figure 2. However, you decide, you should be consistent in your reporting across the manuscript.

3. Figure 7: Immunoblot data were normalized to β-actin prevalence, as many people do. However, there are examples that actin itself is regulated. Therefore, the abundance of actin in both groups must also be stated quantitatively. Otherwise, it cannot be determined to what extent differences in actin prevalence affected the actin-adjusted assessment of the target proteins.

4. Tables 1-2: A) Does it make sense to provide body weight with 2 decimals, i.e., a pretended precision of 10 mg? B) Why are most blood analyte concentrations given in mM, but blood fructose in mg/mL and F1P in ng/mL? Shouldn’t all be given in molar concentrations for consistent reporting? Overall, much of the tables repeats what is already shown in the figures with greater granularity. Is there any reason to show the same data twice?

5. Table 5 refers to an “antagonist group”. However, it remains unclear what this means, as the rest of the manuscript does not refer to such a group.

Experimental design

6. Fructose feeding primarily is not a bladder model but rather a model of hepatic dysfunction induced by metabolic changes. You should not make it sound as if fructose feeding is an OAB model of choice for adult rats (while having been used by some investigators, it is not).

7. L. 79-84: According to this section, the study had two aims: to establish a model of pediatric OAB and to look at short-term fructose exposure. Isn’t this a bit too much within a single two-armed study? Moreover, you wished to investigate “underlying molecular mechanisms”. Validating a model and using it to test biology within the same study creates circular reasoning. A study can either validate a model or it can use an established model to test a biology; you can’t eat the cake twice.

8. Within the model establishment/validation part, it is not clearly stated which (pre-specified!) criteria would make the model useful. Moreover, the Discussion section largely focuses on the biology rather than on the validation of the model.

Validity of the findings

9. As the study is based on the comparison of two diets, both diets must be described specifically. Just stating “standard diet” and “containing 60% fructose” is not enough. The source and exact composition of both diets must be disclosed.

10. Several in vivo tests are mentioned, but a reference explaining how it was done is not provided for any of them. For most tests being reported, I would be unable to repeat them due to a lack of granularity in their description.

11. I am happy to see a statement that randomization was applied, but it is not disclosed how randomization was done.

12. Histological and immuno-histological analyses are notoriously vulnerable to investigator bias. Please state explicitly whether the person scoring the slides was blinded to group allocation. It is also not stated at which magnification the slides were viewed and how many viewing fields were counted per animal.

13. I could not understand the data analysis section in its present form. Does this relate only to the metabolomics part of the study?

14. Statistical analysis: As the study is exploratory by nature, calculated p-values cannot be interpreted as hypothesis-testing but only as descriptive. This should be stated explicitly. Of note, based on a limited number of biological replicates (rats), a large number of parameters are being compared; there is no adjustment of the p-values for such a large number of parameters being tested on the same biological replicates.

15. The overall reporting of results focuses too much on p-values and too little on the biological relevance of findings 1. Without switching back and forth to the tables and figures, we do not get an impression of whether the observed changes are small or large.

16. Based on Figure 2 and l. 317-319, fluid intake was increased, and micturition frequency was increased. Contrary to what you imply, this is not evidence for OAB-like symptoms: if fluid intake goes up, micturition frequency, of course, also goes up – irrespective of whether bladder function is changed or not.

17. The description of statistical tests being used and the multiple inter-group comparisons in the figures do not match intuitively. Thus, it was unclear to me which statistical tests were applied to which inter-group comparisons. Moreover, I do not understand the rationale for calculated p-values for individual time points within an OGTT.

18. Neither Methods nor Results presents a rationale why compound C/dorsomorphin has been tested.

Additional comments

19. L. 14 and 37: If you define OAB as a condition, it should be “overactive bladder syndrome”.

20. I found the results part of the Abstract not to be very informative. It should provide effect size estimators for the 2-3 parameters you consider most relevant.

21. L. 39: Xing 2020 is an inappropriate reference here because it does not provide a definition of OAB. The original ICS definition reported by Abrams 2002 should be cited here.

22. The 1st paragraph of the main Introduction is partly misleading. You talk initially about OAB in general, but without identifying it, you apparently switch specifically to pediatric OAB. For instance, (l. 48) oxybutynin is listed as a first-line approach. This may be true for children, but clearly not for adults because oxybutynin (at least in its IR formulation) has the worst efficacy/tolerability ratio among all anti-muscarinic receptor antagonists. A comprehensive review of existing evidence in adults and children has been published recently 2. Moreover, recent guidelines for OAB in general list antimuscarinics and β3-adrenoceptor agonists as equally being first-line medical treatment.

23. L. 53-75: This paragraph should more clearly separate a) animal from clinical studies and b) what has been suggested vs. what has been proven experimentally or clinically.

24. L. 77: There is no such thing as “neurogenic OAB”. As you correctly state in l. 38-39, the OAB definition by the ICS excludes symptoms due to infection or neurological disorders.

25. L. 76-78: I find the list of OAB animal models to be very selective, focusing only on metabolic factors and outlet obstruction.

26. L. 93: Do you mean the original ARRIVE guidelines or ARRIVE 2.0, which has superseded them several years ago?

27. RRIDs should be mentioned for all animals, kits, antibodies, etc., for unequivocal identification of research resources. Where unavailable, the catalog numbers should be provided.

28. Centrifugation forces should be reported in multiples of g, not as rpm.

29. The section l. 294-315 switches between measurements in blood and those in bladder tissue. This is confusing and should be avoided.

30. The rationale for applying the open-field test (a test frequently used in anxiety and depression research) is not explained.

31. Some abbreviations are introduced without being used thereafter.

32. L. 412-421: This paragraph within Results does not present any results but provides discussion. It should be moved to the Discussion section.

33. L. 568: Apparent use of an AI product is disclosed. However, we are not told for which purpose and which part of the manuscript it was used.

References
1 Amrhein V et al. Nature 2019; 567: 305-307
2 Michel MC et al. Pharmacol Rev 2023; 75: 554-674

---

## Round 0.2 · accepted · Accept

Both reviewers found it acceptable for publication. Currently, some typos in the manuscript, as identified by reviewers, can be corrected during the proofreading stage.

Reviewer 1 ·

Basic reporting

The authors implemented substantial revisions to the article, fully addressing all recommendations. The updated reporting enhanced the reproducibility of the methods used. They incorporated references to the methodologies to support the evaluation of the sources, as well as citations within the discussion section, which strengthened the overall structure of the paper.

Experimental design

No comment.

Validity of the findings

No comments.

Additional comments

Check line 122 there is a repeated word: "Both groups were were fed their respective diets for one week."

Reviewer 2 ·

Basic reporting

.

Experimental design

.

Validity of the findings

.

Additional comments

I have looked at the revised manuscript and find that all of my previous comments have been addressed adequately. Thus, I now endorse publication of the manuscript.